# Ozone in the boreal forest in the Alberta oil sands region

Xuanyi Zhang[1], Mark Gordon[1], Paul A. Makar[2], Timothy Jiang[1*], Jonathan Davies[2], David Tarasick[2]

[1]Earth and Space Science, York University, Toronto, M3J 1P3, Canada
[2]Air Quality Research Department, Environment and Climate Change Canada, Toronto, M3H 5T4, Canada
[*]Now at School of Environmental Sciences, Guelph University, Guelph, N1G 2W1, Canada

*Correspondence to*: Mark Gordon (mgordon@yorku.ca)

**Abstract.** Measurements of ozone were made using an instrumented tower and a tethersonde located in a forested region surrounded by oil sands production facilities in the Athabasca Oil Sands Region (AOSR). Our observations and modelling show that the concentration of ozone was modified by vertical mixing, photochemical reactions, and surface dry deposition.

Measurements on the tower demonstrated that when winds are from the direction of anthropogenic emissions from oil sand extraction and processing facilities, there is no significant increase in ozone mixing ratio compared to when winds are from the direction of undisturbed forest. This suggests that ozone is destroyed by reaction with $NO_x$ from oil sands extraction operations (as well as NO resulting from photolysis of $NO_2$). Vertical gradients of ozone mixing ratio with height were observed using instruments on a tethered balloon (up to a height of 300 m) as well as a pulley system and 2-point gradients

within the canopy. Strong gradients (ozone increasing with height near 0.35 ppb $m^{-1}$) were measured in the canopy in the evening and overnight, while morning and daytime gradients were weaker and highly variable. A 1D canopy model was used to simulate the diurnal variation of the in-canopy gradient. Model results suggest an ozone dry deposition velocity of between 0.2 cm $s^{-1}$ and 0.4 cm $s^{-1}$ for this location. Sensitivity simulations using the model suggest the local NO concentration profile and coefficients of vertical diffusivity have a significant influence on the $O_3$ concentrations and profiles

in the region.

## 1 Introduction

Canada's largest oil sands deposits areas are found in the Athabasca Oil Sands Region (AOSR) of northern Alberta. The increasing oil sands production has led to increased environmental concern for the nearby forest ecosystem (Li et al., 2017). The processes of oil and gas extraction from oil sands include surface mining to turn surface oil sands into crude oil, well

injection to pump deeper bitumen onto the surface, extraction of bitumen from oil sands with the water-based process and the upgrading of bitumen into hydrocarbon streams (Natural Resources Canada, 2016).

Ozone is a photochemical pollutant in the troposphere. It is produced there by the photochemical oxidation of carbon monoxide, methane, and non-methane volatile organic compounds in the presence of nitric oxide (NO) and nitrogen dioxide ($NO_2$). All these species are emitted from activities in the oil sands.

In the AOSR, Cho et al. (2017) observed no statistically significant increase in long-term ozone measurements over the 1998 to 2012 period, despite an 8% annual increase in $NO_x$ emissions. The lack of a local ozone increase was attributed to $NO_x$ from local emissions, which results in titration of ambient ozone. Similarly, Aggarwal et al. (2018) measured ozone levels in the Alberta oil sands region (between the ground and a height of 1.8 km) which were lower than or equal to the background ozone mixing ratio, which is also attributed to NO titration. The absence of enhanced ozone downwind of industry was

associated with air temperatures less than 20ºC and vertical mixing of polluted and clear background air.

The motivation for this study is to a) determine how pollutant emissions associated with oil sands extraction modify ozone concentration in the surrounding forest, and b) estimate the dry deposition velocity of ozone to the surrounding forest. A boreal forest site was chosen that is surrounded by oil sands processing facilities including those operated by Syncrude, Suncor, Canadian Natural Resources Limited (CNRL), as well as other facilities. Since exposure to ozone reduces

photosynthesis, growth, and other plant functions (Felzer et al., 2007), we investigate what effect the elevated pollution levels of the AOSR have on the surrounding boreal forest and determine the rate of ozone uptake to the forest. The importance of correctly modeling dry deposition to the forest is demonstrated by Clifton et al. (2020b), who find that variation in deposition schemes leads to mean summertime biases of −4 to 7 ppb. A review of ozone deposition velocity schemes used in current models may be found in Clifton et al (2023).

Makar et al. (2017), herein M17, demonstrated that the turbulence and shading effects of forests on ozone mixing and chemistry have been poorly modeled in global and regional air-quality models. They found that including both these effects in a regional air-quality model accounted for 97% of the previous positive bias in forested regions. Approximately one-third of this improvement was attributed to the shading effect, while two-thirds was due to the change in turbulence parameterization. Hence, this paper suggests that any accurate modeling of ozone within a forest must include both

turbulence and shading effects. Testing currently underway with the CMAQ air-quality model supports these results, showing a significant improvement in model surface ozone biases when these effects are included (Campbell et al, 2021).

Clifton et al., 2021 used a large eddy simulation coupled to a multilayer canopy model to investigate ozone removal by a deciduous forest. They found that organized turbulence leads to heterogenous mixing which can slow down or speed up reaction rates of ozone at different heights in the canopy. They found low covariance between ozone mixing ratio and leaf

uptake (due to the effects of organized turbulence). This finding effectively questions the use of a deposition velocity in estimating ozone fluxes since the uptake flux of ozone is equal to the product of ozone mixing ratio and deposition velocity. Nevertheless, the analysis also suggests that organized turbulence does not likely bias estimates of ozone dry deposition during summertime afternoon conditions.

Finco et al. (2018) analyzed ozone deposition in a deciduous forest (in a highly polluted region in Italy) based on

measurements from the understory up to above the canopy. They found that ozone deposition increased with height within the canopy. Ozone fluxes were much higher at the canopy level within the foliage (24 m) compared to the above canopy region, and stomatal deposition was the main ozone removal process (with less than 20% removed by chemical reaction with NO and by non-stomatal deposition). High stomatal conductance leads to high stomatal uptake of ozone. The peak of

stomatal conductance occurs when the ambient air is warm and humid (Ducker et al., 2018). Stomatal closure occurs in the

nighttime, and then non-stomatal dry deposition processes become the main ozone deposition process (Pilegaard, 2001). The dominant chemical loss process is NO reaction with $O_3$ below the canopy (Kaplan et al., 1988). There are also monodirectional fluxes of NO and $NO_2$, with NO emitted from the soil and $NO_2$ deposited to the ground (Finco et al., 2018). However, Wolfe et al. (2011) have demonstrated that chemical loss due to VOCs can also be significant.

Rannik et al. (2012) measured ozone dry deposition to a boreal forest using eddy covariance over a 10-year period. They

determined that the ozone deposition velocity ($v_d$) was 0.4 cm s$^{-1}$ during the peak growing season. Multivariate analysis demonstrated that $v_d$ was correlated with photosynthetic capacity of the canopy, vapour pressure deficit, photosynthetically active radiation, and monoterpene concentration. Wu et al. (2016) determined $v_d$ at a mixed boreal-temperate transition forest over a 5-year period using a gradient approach. The highest monthly mean of ozone deposition velocity was 0.68 cm s$^{-1}$ and the 5-year average and median were 0.35 and 0.27 cm s$^{-1}$ respectively. The average ozone deposition velocity in the

summer was near 0.4 cm s$^{-1}$ at night, increased to 1 cm s$^{-1}$ in the morning, and then decreased steadily through the day back to the nighttime value.

A recent review of ozone deposition by Clifton et al. (2020a) highlights the need for both short-term field intensives and long-term deposition sites. The review synthetizes the current knowledge of deposition pathways, including stomatal, non-stomatal, and soil uptake and in-canopy chemistry. While our study is not able to distinguish these various pathways, the

motivation is to investigate how oil sand extraction and processing affects ozone mixing ratios and to determine the total dry deposition velocity at this location. This study focusses on ozone deposition analysis using vertical ozone mixing ratio gradients within and above a jack pine forest canopy in a forest region surrounded by oil sand production facilities. Ozone mixing ratio was measured above and within the forest canopy, as well as ultraviolet radiation (UV), photosynthetically active radiation (PAR), and meteorological variables. Gradient and mixing ratio measurements are compared with a 1D

canopy model with various dry deposition schemes using a "big leaf" approach, which assumes a deposition flux at the lowest model layer. This paper is a companion paper to Jiang et al. (2022) and Gordon et al. (2022) which respectively investigate aerosol and $SO_2$ dry deposition at this site.

## 2 Methods

### 2.1 Site Location and Instrumentation

The study site is characterized by a surrounding homogeneous jack pine forest with flat topography, in turn surrounded by oil processing and extraction facilities. This selected forest area (Fig. 1) is far from cities, and the nearest habitation (Fort McKay) is about 11 km to the Northwest of the study site, with a population of only 700. The nearest town of Fort McMurray is approximately 45 km south of the site. The nearest highway is approximately 650 m south of the study site. The traffic on this highway is light, with usually 1 or 2 transport trailers or large trucks per minute. Thus, the effect from the

town and the highway can be assumed to be negligible for this study and the pollution detected in the forest was mainly from the surrounding oil sands processing facility plumes.

Emissions in the region are summarized in Zhang et al. (2018), in which national, provincial, and local emissions inventories for the Oil Sands Region between 2010 and 2013 (up to 7 years prior to the start of this study) are reviewed. Zhang et al. (2018) report annual totals of 18,000 t (tonnes) CO, 39,600 t $NO_x$, 1,000 t $PM_{2.5}$, 1,100 t $PM_{10}$, 760 t $SO_2$, and 34,000 t VOCs. More than 40% of the CO, $PM_{2.5}$, and $PM_{10}$, emissions were from the Suncor facility (Fig. 1), while nearly 50% of the $SO_2$ and VOC emissions were from the Syncrude facility (Fig. 1). Significant VOCs (> 1000 t/a) included higher alkenes, higher alkanes, higher aromatics, propane, isoprene, and toluene.

In July 2017, the York Athabasca Jack Pine (YAJP) tower was installed at the study site at 57.1225 N 111.4264 W. The tower is 29 m high, and the canopy is about 19 m high (with the tallest trees ranging from 16 m to 21 m in height). In the 2017 field study, an ozone analyzer (Model 205, 2B Tech) and a UV sensor (CUV5, Kipp & Zonen) were mounted on a sub-canopy tower pulley system to measure ozone and UV profiles between the ground and a height of 16 m. A table platform was used on the pulley (with ropes attached at each corner) to ensure the UV sensor remained level. The data were collected for periods of 5-min intervals at 5-m height intervals after ensuring the sensors were level and not moving.

During a 2018 summer intensive field campaign (9 to 17 June) two ozone monitors (Model 205, 2B Tech.) were mounted on the tower at heights of 25 m and 2 m. Ozone and $SO_2$ analyzers (49i and 43i, Thermo Scientific) sampled from a height of 2 m. A generator was used to power the instruments during the field intensives, which was placed 100 m from the tower in the northeast direction (at a wind direction of 40º), since regional winds are typically not from this direction. For 3 days during this study, a Vaisala Tethered-Balloon system (Vaisala DigiCORA) was used for short-term ozone profile measurements. The balloon lifted one tethersonde (TTS111, Vaisala) and an ozonesonde (Smit et al., 2007) with an Arduino data logger. Calibration and uncertainty of the ozone instruments is discussed in the following section.

After the 2018 field experiment, a solar-powered ozone monitor (Model 405, 2B) was left on the tower at a height near 19 m. This ran continuously until mid-July, and then intermittently until mid-August, after which there was inadequate sunlight for the solar-power system. In March 2019, two solar-powered ozone monitors (Model 405, 2B) were left on the tower at heights of 2 m and 25 m for long-term monitoring. The monitors were activated 4 times per day (02:00, 08:00, 14:00, and 20:00 local) for one hour duration. The sampling frequency during these time periods was 0.25 Hz. To allow the instrument to equilibrate after each cold start, the measurement for each period was taken as the average value of the last 15-minutes of measurement in each hour. The uncertainty associated with this technique is discussed in the following section. These monitors remained operational until late June 2019.

During August 2021, there was a third summer intensive study at the tower. No ozone measurements were made during this field study, but $SO_2$ measurements (43i, Thermo Scientific and AF22e, Envea) at ground level and a height of 30 m and size-resolved sub-micron aerosol measurements (UHSAS, DMT) helped to further identify wind sectors bringing polluted air to the site. The $SO_2$ and aerosols measurements are discussed in our companion papers Gordon et al. (2022) and Jiang et al. (2022) respectively.

Permanent instruments on the YAJP tower (solar powered) used for the following analysis include sonic anemometers (ATI Inc.) at heights of 29 m and 5.5 m, photosynthetically active radiation (PAR, LI-190, Licor Inc) measured at heights of 29 m, 15.9 m, and 2 m, and a gas analyzer ($CO_2$/$H_2O$, LI-7500, Licor Inc) at a height of 29 m. These instruments were factory calibrated and data were quality controlled through visual inspection of the time series, resulting in rejection of less than 0.1% of the data.

The Wood Buffalo Environmental Association (WBEA) operates a meteorological tower identified as "1004" which is approximately 540 m south of the YAJP tower. The 1004 tower measures hourly values of: air temperature, relative humidity (RH) and winds at heights of 2, 16, 21, and 29 m; PAR at heights of 2, 16, and 21 m; and atmospheric pressure (at 2 m).

For chemical species not measured in this study (NO, $NO_2$, and eddy diffusivity $K$), we use output from the GEM-MACH model (Global Environmental Multiscale-Modeling Air-Quality and Chemistry), which is the regional chemical transport model used by Environment and Climate Change Canada. GEM-MACH has been used for numerous modeling studies focussed on the AOSR (e.g. Makar et al., 2018; Whaley et al., 2018; Fathi et al., 2021). The model provides turbulence parameters which are consistent with meteorological forecasts for the region. The GEM-MACH resolution is 2.5 km and the mines and upgrading facilities are more than 10 km (~4 grid squares) from the tower location. Hence, GEM-MACH can resolve source locations within at least $\pm$ 7°.

## 2.2 Ozone Instrument Uncertainty

The 49i analyzer was laboratory-calibrated prior to the study (a 7-point calibration up to 120 ppb with $R^2 = 0.997$). The standard deviation in the 5-second measurement during calibration was 1.9 ppb. All Model 205 and ozonesonde monitors were calibrated in the field against the 49i analyzer by running the instruments side-by-side for 9 consecutive days period (at a 0.5 Hz frequency) prior to the long-term averaging periods. The ozone mixing ratio varied from near 1.3 ppb to 38 ppb during this period. The root-mean-square errors (RMSE) against the calibrated 49i were less than 2 ppb (at 0.5 Hz). The 2B Ozone Monitor specifications (2B Specifications) give a drift value of $< 1$ ppb day$^{-1}$ and $< 3$ ppb year$^{-1}$. Over the 9-day period, the RMSE (at 0.5 Hz) showed no discernable trend with time. A least-squares fit of RMSE with time give a trend of 0.004 ppb day$^{-1}$ (which is not significantly different from zero at a 95% confidence level (C.I.)). Hence, we assume minimal drift during the 3-month measurement period. Water vapour interference is assumed to be minimal since the 2B analyzers have a built-in dryer and heater to eliminate water vapour interference and temperature effects (2B Specifications), and the inlet tubing is only 10 cm in length.

During the long-term, 2019 measurements, when the monitors were activated 4 times per day for one-hour durations, the monitors were allowed to stabilize for 45 minutes and only the last 15-minutes of measurements were used. The manufacturer specifies (2B Specification) a 20-minute warm-up period. The stabilization of the instrument is demonstrated in the supporting information (Fig. S1) from the measured data, indicating full stabilization may require approximately 35 minutes. A truncated mean is calculated from the last 15 minutes, with outliers more than 3 standard deviations from the

mean removed (resulting is removal of less than 0.5% of data). The average standard deviation in this 15-minute interval is 1.8 ppb, which gives a 95% C.I. of $\pm 0.24$ ppb (for each 15-minute average).

## 2.3 Ozone Modeling

To model photochemical processes, vertical turbulent mixing, and deposition in the canopy, we use a one-dimensional (1D) canopy model created by Makar et al. (1999). The model has 1001 levels in the vertical direction, and each level has 1-m spacing. The model uses 30-minute interval input data. It includes 268 chemical reactions associated with 79 output species. In the 1-D canopy model, the rate of change of each chemical species mixing ratio ($C$) at each model level is calculated due to their emissions or uptake (positive or negative $E$, respectively), chemical reactions ($f$) and turbulent mixing (Eq. 1) at

each layer. In Equation 1, subscript m represents different chemical species, subscript n represents the vertical layer, and $K$ is the eddy diffusivity.

$$\frac{\partial C_{mn}}{\partial t} = E_{mn} + f_{mn} + \frac{\partial}{\partial z}\left(K_n \frac{\partial C_{mn}}{\partial z}\right), \tag{1}$$

Further modifications were made to the model outlined in Stroud et al. (2005) and Gordon et al. (2014) to include sesquiterpenes, modify the turbulent mixing code, and include surface dry deposition. Deposition is added as uptake (a

negative mass rate of change, $E$) at the lowest level. This model version uses operator splitting in each minute. First, each species diffuses for 30 seconds (with a time step of 1 second) using a Crank-Nicholson numerical scheme to solve the turbulent mixing term in Eq. 1. This is followed by 1 minute of uptake or emissions ($E$) and chemistry ($f$). Then the species diffuse for another 30 seconds. The operator splitting process is repeated 30 times for each 30-min output time step.

Model input variables are updated every 30 minutes. The model uses input data of pressure, PAR, and RH at a single height.

Air temperature and NO are input for the lowest 50 levels with 1 m spacing and the turbulent eddy diffusivity ($K$) is input for all 1001 levels at 1 m spacing. Air temperature profiles were linearly interpolated using the 2, 16, 21, and 29 m measurements from the 1004 tower. The air temperature above a height of 29 m was modeled assuming a dry adiabatic lapse rate (0.0098 K m$^{-1}$) above this height. As a sensitivity test, we run versions of the model with constant air temperature above 29 m and with a dry adiabatic lapse rate above this height (Supporting Information S1). Pressure, PAR, and relative humidity

are required as model inputs at the canopy height. These variables were measured using the gas analyzer (LI-7500) and PAR (LI-190) sensors. Based on the availability of driving data, we ran the model for 6.5 days from 18:00 (local time) on 20 Jun to 06:00 on 27 Jun 2018. NO was not measured at the site, and the choice of NO inputs as a model boundary condition is discussed in the following section. The measured ozone mixing ratio at a height of 22 m was used to initiate the model. The first 12 hours of simulation are not used in this modeling analysis to allow for model spin-up, which gives 6 days of

measurement-to-model comparison.

Above the canopy height, the eddy diffusivity, $K_n$, was extracted from the GEM-MACH model for the simulated time period at the 16 GEM-MACH vertical levels ranging from approximately 21 m and 1040 m (a.s.l.) in 1-hour intervals. Half-hourly

$K$ values, required as inputs for the 1D canopy model, were linearly interpolated form the hourly values. The eddy diffusivity is calculated in GEM-MACH as (Mailhot and Benoit, 1982)

$$K = 0.516 \, \kappa \, z_m \, \frac{\sqrt{e}}{\phi} \tag{2}$$

where $\kappa = 0.4$, $e$ is the turbulent kinetic energy, and $\phi$ is a stability parameter. Since $e$ was measured at the YAJP tower at a height of $z_m = 29$ m, the GEM-MACH profiles are corrected to this value using a time-varying correction ratio (applied equally to all heights). Hence, the GEM-MACH $K$ vertical profile shape is preserved, but the modeled values were adjusted to observations. The stability factor and Obukhov length are determined following Garratt (1994) as

$$\phi = \begin{cases} 0.74 \left(1 - 9\frac{z_m}{L}\right)^{-1/2} & \frac{z_m}{L} < 0 \\ 0.74 + 4.7\frac{z_m}{L} & \frac{z_m}{L} \geq 0 \end{cases}, \tag{3a}$$

$$L = -\frac{u_*^3 \, T}{\kappa \, g \, \overline{w'T'}}, \tag{3b}$$

using the friction velocity ($u_*$), air temperature ($T$), and heat flux ($\overline{w'T'}$) measured at a height of $z_m = 29$ m on the YAJP tower (and $g = 9.8$ m s$^{-2}$).

Within the canopy, $K$ is modeled following the parameterization outlined in M17, where a generic within-canopy profile of 
$K$ is imposed and is normalized relative to an above-canopy $K$ value, with the intent of capturing the typical "shelf" in the decrease of $K$ with decreasing height seen in multiple forest observations (e.g. Raupach, 1996). The M17 parameterization is normalized to the GEM-MACH model value of $K(z_l)$ at the lowest layer height, $z_l$, which was approximately 50 m in M17. Here, we normalize to the lowest model layer height in the updated version of GEM-MACH, which is $z_l = 23$ m. Since this is only slightly higher than the canopy height of $h_c = 19$ m and the effect of canopy turbulence has been shown to extend to 
twice the canopy height (Mölder et al., 1999), we also normalize to the height of the second lowest model layer $z_l = 42$ m (from M17) as a sensitivity test (Supporting Information S1). When normalized to the GEM-MACH $K(z_l)$, the M17 parameterization of $K$ within the canopy is only a function of stability ($h_c/L$).

Ozone deposition was modeled with a "big leaf" assumption, where deposition is entirely attributed to the surface ($z = 0$) layer and uptake to the canopy (such as deposition to pine-needle surface or stomata) is ignored. This is modeled as a 
surface-flux boundary condition in the turbulent mixing solver. While a vertical distribution of uptake (or locating the "big leaf" at a specified height above the surface) would be more realistic, this would require placement of the ozone uptake in the emission and chemistry operator step (as opposed to the turbulent mixing operator).

Since ozone is depleted in the 1D model through deposition and chemistry, it must be replaced at the upper boundary of the model (1001 m). This is done by holding the concentration at the highest model layer constant as an upper boundary 
condition. This value is estimated based on a measured peak daily ozone near 50 ppb (at a height of 22 m) and a continued increase of ozone with height of 0.01 ppb m$^{-1}$ (based on measurements shown in Sections 3.2 and 3.3), giving a constant

upper-layer ozone value of 60 ppb. The choice of model height (1001 m) was determined by inspecting vertical ozone profiles from ozonesonde launches at Bratt's Lake (Astitha et al., 2018), which is located approximately 500 km SSW of the oil sands region. The aggregate vertical profile shows a consistently steep gradient between the surface and a height of 1 km

(approximately 20 ppb km$^{-1}$) and a much weather gradient between 1 km and 2 km (< 3 ppb km$^{-1}$). Based on this, we choose a 1 km upper boundary of the model and ozone is held constant at this height. Sensitivity to both the assumed constant value and the choice of model height (1001 m) are tested in supporting information (S1).

Canopy shading is accounted for in the model with PAR attenuation through the canopy as (Makar et al., 1999)

$$I_i = I_o \exp(-k \sum L_i / \cos \theta), \tag{4}$$

where $I_i$ is PAR at each level $i$, $I_0$ is the above canopy PAR, $\sum L_i$ is the leaf-area index (LAI), summed from the canopy top to level $i$ within the canopy, $k$ is an extinction coefficient, and $\theta$ is the solar zenith angle. The measured PAR above the canopy and at the surface suggests a value of $k = 0.68$, which is close to the value of 0.70 used by Stroud et al. (2005) for a pine plantation in North Carolina, but much higher than the value of 0.31 for jack pine determined from the classification scheme described in M17 (with $k = G\Omega = 0.5 \times 0.62$). Sensitivity of the model to this parameter is also tested in supporting

information (S1).

The canopy LAI profile was measured by analyzing fish-eye lens video at various heights (mounted on the pulley system) with the Gap Light Analyzer (GLA) software (Frazer et al., 1999). The LAI profile determined for the forest is shown in Figure 2. The total LAI was measured using ground level images from the area in the vicinity of the tower. This gives a total LAI of 1.17 at the site. We compare this measured value to LAI from MODIS-derived seasonal LAI maps at 2.5 km

resolution (Zhang et al, 2021). The site location is near the edge of two 2.5 km grid cells (see supporting Figure S2) with values of 1.09 and 1.27 in January and 1.89 and 2.23 in July. While the site is in an area dominated by jack pines, the surrounding area (within a few kilometers) also includes black spruce dominated stands and sphagnum dominated muskeg. The seasonal variation of the MODIS-derived values suggests deciduous trees in the surrounding area. To investigate a possible underestimation of the surrounding representative LAI, we ran two sensitivity tests with LAI values of 2 and 3.5.

While the modified LAI affects the light penetration into the canopy, it does not affect the turbulence profile, as the M17 parameterization of $K$ does not include dependence on LAI.

As is outlined in Makar et al. (1999), the model includes forest emissions of isoprene and monoterpenes following Guenther et al. (1993). The emission rates are functions of air temperature (used as a proxy for leaf temperature) and LAI, and emissions are at each model layer in the canopy. Here we use base emission rates of 8 µg g$^{-1}$ h$^{-1}$ for isoprene and 2.4 µg g$^{-1}$ h$^{-1}$

$^{1}$ for monoterpene from Guenther et al. (1995) for boreal conifers (which includes both the local jack pine forest and surrounding black spruce). Sesquiterpene emission rates are set to 1/3 the monoterpene emission rates following Stroud et al. (2005). We test the sensitivity to these emission rates in the supporting information (S1).

A series of model configurations were chosen to investigate different physical mechanisms and their potential effect on the diurnal variation of ozone mixing ratios and the gradients above and within the canopy and to improve the measurement to

model comparison. These model configurations are listed in Table 1. The model was run for each configuration for the period from 18:00 (local) 20 June to 06:00 (local) 27 July 2018. We disregard the first 12 hours for model spin-up, resulting in 6 days of model output. The first 5 configurations are variations in input NO, discussed in the following section. Configurations #6-8 vary the ozone deposition velocity to 0 (#6), 0.2 cm s$^{-1}$ (#7), and 0.8 cm s$^{-1}$ (#8) (from the base case of 0.4 cm s$^{-1}$). Although it is unrealistic to assume no deposition of ozone, this configuration was included as a demonstration of the extent to which the gradient depends on deposition alone. Configurations #9 and #10 vary the strength of turbulent mixing by a factor of 0.5 and 2 respectively (at all heights). To compare model output and measurements, the 10-min measurements at a height of 22 m were averaged to 30-min values.

## 2.4 NO Simulations in the Model

Since NO is an advected species, carried to the site from upwind emissions, it is prescribed as an input variable at each time-step and is not modeled as a time-varying species according to Equation 1. Initial model runs demonstrated that the model's ability to predict ozone is strongly dependent on the choice of NO used as a boundary condition. Due to this sensitivity, we consider and compare several different NO input scenarios to determine the effect on ozone mixing ratios and gradients. These scenarios comprise of an optimized constant NO value (config. #1), NO modeled from GEM-MACH output (configuration #2 in Table 1), NO as a function of wind direction (config. #3), and elevated NO near the surface (configs. #4 and #5).

Firstly, surface-level NO mixing ratios were extracted from the GEM-MACH model (M17) in 1-hour intervals (which were linearly interpolated to 30-min values). The 1D canopy model assumes a constant NO mixing ratio with height. Initial test runs showed that using the GEM-MACH NO values, the timing of plumes arriving at the YAJP tower is mis-aligned relative to measurements. The proximity of nearby stack sources (within ~16 km) means that small changes in wind direction can determine whether the YAJP tower location is inside or outside the plume in the regional model. Hence, we also run another model case with NO mixing ratio values as a function of measured wind direction, based on GEM-MACH NO and wind direction over the same period. In this analysis the GEM-MACH NO values are binned by modeled wind direction in 18 bins (with 20° width) and the median NO value for each bin is calculated in order to capture NO values when GEM-MACH predicted plumes coming from the sources. These median NO values were then used as input for the 1D canopy model based on the measured wind direction at the tower, thus correcting any plume misalignment caused by small errors in GEM-MACH's forecasted wind direction.

In addition to these NO simulations, we also test the 1D model with constant, time-invariant NO mixing ratios. A set of optimization runs demonstrated that the lowest RMS error of ozone mixing ratio was achieved with a constant value of NO = 0.05 ppb. To demonstrate the model sensitivity to NO, we include tests with 0.01 and 0.1 ppb in the supporting information (S1).

Decaying plant matter can also be a source of NO. Finco et al. (2018) measured NO in a forest near the surface and at 5 heights through the canopy. While the NO between heights of 5 m and 41 m varied from 0.1 to 2.5 ppb in the Finco et al.

study, measurements at a height of 0.15 m ranged from 5 to 20 ppb (Finco et al., 2018). Using the parameterization of NO surface emissions from the GEM-MACH model (Williams et al., 1992) with the temperatures over the modeling period

suggests that between 0.5 and 1 ppb of NO should be added to the bottom layer of the model during each 30-min time step. While these NO emissions would realistically be diffused upward into the canopy and the profile would depend on turbulence and stability, we approximate these emissions with two case studies where the NO levels in the 3 lowest 1-m model layers are held at either 1 or 5 ppb (with NO = 0.05 ppb for heights above 3 m).

## 3 Results and Discussion

### 3.1 Diurnal Ozone Variation by Wind Sector

Figure 3a demonstrates the variation of measured $SO_2$, $CO_2$, and sub-micron aerosol total number ($N$) measured at the tower site with wind direction. The $SO_2$ measurements (reproduced from Gordon et al., 2023) are from two time periods (9-19 June 2018 and 7-25 August 2021). $SO_2$ is primarily associated with large stack emissions, whereas ozone precursors such as $NO_x$ can also be emitted from vehicles and machinery (e.g. Zhang et al., 2018). Lacking precursor measurements in this study, we

rely on $SO_2$, $CO_2$, $N$, and the area satellite map to differentiate sectors based on pollutant types. The elevated $SO_2$ mixing ratios between 160º and 250º demonstrate polluted air being transported to the site, likely from either the Suncor (13.5 km at 195º) or the Syncrude (18 km at 225º) processing facilities. Sub-micron aerosol number ($N$) and $CO_2$ measurements demonstrate a similar enhancement in the polluted wind sector, but also show enhancement from the north (approximately 0 – 40º for $N$ and 340 – 45º for $CO_2$). Based on these measurements, we very broadly define three wind direction sectors with

*polluted* air (i.e. enhanced $SO_2$, $CO_2$, and $N$) from 160 – 250º, open *forest* (background $SO_2$, $CO_2$, and $N$) from 40 – 160º, and *other* (no enhanced $SO_2$, but varied industrial sources with some $CO_2$ and $N$ enhancement within the sector) from 250º – 40º. These wind sectors are shown in Figure 1.

NO and $NO_2$ are included in the comparison using GEM-MACH output for the period between 1 Jun and 17 Aug (Fig. 3b). The GEM-MACH $SO_2$ output demonstrates a similar pattern to the measurements, with elevated values when winds are from

the *polluted* sector, although there are a few elevated values when winds are from the *Forest* and *Other* sectors. In the model output, these represent cases where wind changes in the model result in $SO_2$ plume directions "looping" so that plumes originating at emissions sources arrive from the other directions. Some emissions form the *other* sector may originate at more distant facility sources between Fort McKay and Bitumount (Fig. 1). Five such cases where GEM-MACH predicted these events arriving from anomalous directions may be seen in Figure 3b. Both NO and $NO_2$ model outputs demonstrate

patterns similar to the $CO_2$ and $N$ measurements, with elevated values when winds are from the *polluted* sector, low values from the *forest* sector, and a mix of low and elevated values from the *other* sector. We note that the delineation between the *forest* and *polluted* sectors (at 160º) is not as clear in the GEM-MACH $NO_x$ output compared to the $SO_2$ measurement and model output; however, tests demonstrated that moving this line to 145º (for example) had little effect on the results described herein. We also note that the GEM-MACH $NO_2$ to NO ratios indicate that the $NO_x$ arriving at the site from

"forest" and "other" directions is significantly more aged (much higher $NO_2$ to NO ratios) than $NO_x$ arriving from the "polluted" direction. This indicates that the $NO_x$ from the polluted direction is relatively fresh, while that from the other directions has experienced more significant photochemical aging, likely due to more distant sources or eventual recirculation of oil sands emissions.

While our companion paper (Jiang et al., 2022) describes source locations for aerosols with finer angular resolution, this is
not possible with the ozone measurements since ozone mixing ratio varies by time of day. There are not enough data to separate both wind direction and time-of-day into more than 3 sectors. Diurnal profiles of ozone measurements separated by the 3 sectors are shown in Figure 4. For ozone measurements made while the generator was operational, measurements from $40 - 60°$ are removed from the open forest sector (but these angles are included for measurements during solar powered operation). Truncated means (within 3 standard deviations) are shown with shaded areas showing 95% C.I., which
demonstrates the significance of the differences between sectors. All sectors show similar temporal patterns. In the spring and summer months ozone mixing ratio is highest in the late afternoon and is lowest just before sunrise (which ranges from 04:30 to 07:00 for the date ranges shown).

Winds are predominantly from the SW in the region, so there are limited data from the *forest* sector, especially for the 1 week of data from Jun 2017. Generally, the ozone mixing ratio was highest when winds were from the *forest* sector,
although this is not the case through the afternoon/evening for the longer period 2018 (2 months) measurements, where ozone is lower from the *forest* compared to the *polluted* sector. In 2017 and the longer period (2 months) in 2018, the late afternoon and evening ozone levels when winds are from the *polluted* sector are higher than the *other* (primarily industrial but not forest) sector. The longer period measurements in 2018 (2 months) and 2019 (~3 months) demonstrate elevated overnight ozone levels transported from the *forest* sector relative to the *other* sector, likely representing background air
unmodified by oil sands emissions, although these values are not statistically different from the *polluted* sector (as demonstrated by the overlap of the 95% CI in Figure 4).

Hence, the long-term diurnal averages (separated by sector) suggest no significant ozone increases associated with industrial pollution, while short-term summertime ozone data shows inconsistent results, with either higher or lower ozone from the industrial sector. The impact of tropospheric folding events, known as stratospheric intrusions, can impact ozone mixing
ratios at the surface (Pendlebury et al. 2018). Other work (Makar et al, 2023) shows a high correlation between monthly ozone averages and the number of stratospheric ozone exchange events occurring within each month, the latter detected by ozone LIDAR within 20 km of the site (Makar et al, 2023). These events have been shown to contribute an additional 10 ppbv to monthly average ozone relative to the ambient atmosphere prior to the events. Since these events happen at varying frequencies with time scales on the order of 1 week, they provide a likely cause of higher ozone over short periods, while
over longer periods the effects of the intrusions would be averaged out. Lidar measurements outlined in Makar et al. (2023) demonstrate a stratospheric intrusion in the AOSR on 6 to 7 Jun 2018, which likely modified the ozone mixing ratios during the 10-day 9 to 18 Jun 2018 period, resulting in more variability and higher concentrations relative to the longer measurement periods. Since intrusion frequencies are relatively constant between Jan to Jun (and less frequent in late

summer), we do not expect the time intervals of the different measurement periods to have a significant effect on the ozone differences between sectors.

These results are similar to previous studies that demonstrate no significant increase in ozone levels with increasing oil sands development (Cho et al., 2017) or ozone levels in the vicinity of oil sands production that are equal to or lower than the background levels (Aggarwal et al., 2018), at these distances from the sources. As with both Cho et al. (2017) and Aggarwal et al. (2018), we hypothesise that this it due to ozone titration by NO. Although the results shown here are highly variable, there appears to be no significant increase in ozone related to increased air pollution.

## 3.2 Ozone Vertical Profiles

Understanding the vertical variation of ozone in and above the canopy is necessary since we use the comparison of measured and modeled gradients to infer deposition velocity. Here we describe vertical profile measurements of ozone and compare these measurements to other studies.

Measurements of ozone mixing ratio and UV radiation within the canopy are shown in Figure 5. These measurements were made on the tower pulley system in the summer 2017 campaign. Ozone tended to increase with height when the UV radiation was also increasing with height, which is the same as the Chen et al. (2018) analysis. This gradient could be due to ozone stomatal uptake and/or chemical reactions such as near-surface $NO_x$ titration (Chen et al., 2018; Finco et al., 2018). In many profiles, there is a peak of ozone mixing ratio within the canopy near a height of 4 m; however, in many cases, this peak is within the variability of the measurements. Finco et. al (2018) found that the ozone mixing ratio in the mid-level of the forest canopy is about 2.5% higher than the ozone mixing ratio above the canopy. The shading effect (demonstrated by the UV measurements) is in good agreement with the LAI profile (Fig. 2), where the lowest UV values coincide with the higher LAI value in the lower canopy (between 2 – 8 m), while the shading above 10 m is generally less pronounced.

Longer-term measurements (from 27 March to 23 June 2019) at fixed heights of 2 m and 25 m were used to determine the ozone gradient, as shown in Figure 6. Measurements indicate a stronger gradient overnight (near 0.35 ppb m$^{-1}$) with weaker gradients in the morning (0.18 ppb m$^{-1}$) and afternoon (0.24 ppb m$^{-1}$). As discussed in Section 2.2, the 95% C.I. for each 15-mintute average is 0.24 ppb, which would imply a 95% C.I. in each gradient measurement of 0.02 ppb m$^{-1}$. Based on the variability in the long-term measurements, the 95% C.I. of each of the 4 mean gradients is < 0.03 ppb m$^{-1}$. Hence, all measured gradients are significantly different from zero and the overnight/daytime difference is significant. A strong gradient would be expected during the night due to increased stability and a greater decoupling of the air above and below the canopy (M17). During the daytime, gradients may be affected by both the decreased photolysis rates within the shaded canopy and the increased turbulence due to afternoon convection. The presence of the canopy may reduce mixing relative to an open space. Continued deposition and/or titration of ozone through the night when there is little mixing will create a stronger gradient as ozone is removed within the canopy. Wu et al. (2016) demonstrate a similar diurnal trend of ozone gradient at a mixed temperate forest in summer with gradients near 0.35 ppb m$^{-1}$ overnight and 0.15 ppb m$^{-1}$ through the day. Our modeling efforts in the following section will attempt to replicate this diurnal variation.

The short-term summertime gradient measurements within the canopy (Fig. 5 and red squares in Fig. 6) are much more variable than the other time periods. The gradients are likely sensitive to short-term variation in ozone mixing ratio during the profiles due to changes in wind direction. These short-term gradients were determined using the difference between the highest measurement and the measurement near a height of 3 m. The near surface measurements were not used for this purpose because of the noted ozone peak near a height of 4 m (Fig. 5). The gradient between heights of near 14 m and near 3 m is therefore considered to be a better representation of the in-canopy ozone gradient. Of the 8 short-term gradient measurements (2017), 3 gradients are very similar to the 2019 long-term spring measured gradients (0.14 to 0.32 ppb m$^{-1}$), 3 gradients are near zero (although they are within 1 standard deviation of the long-term spring measurements), and 2 of the profiles show strong negative gradients (ozone mixing ratio decreasing with height). Near-zero gradients could be caused by strong mixing. Negative gradients could be due to a change in air-mass above the canopy with a change in winds bringing cleaner ozone-free air (or, alternatively, plumes of $NO_x$ aloft may decrease ozone aloft, resulting in a decrease of ozone with height). The more frequent positive ozone gradients are consistent with surface-based ozone loss, due to deposition and/or surface-based chemical losses.

The ozone vertical profiles measured by the tethered-balloon system in 2018 on three measurement days are shown in Figure 7. The measurement was up to a height of 300 m. Each value shown in the figures is an average over a 20-m interval, and the error bar at each is the standard deviation. The gradient determined from the difference between the highest (300 m) and lowest (20 m) averages ranges from −0.0014 to 0.02 ppb m$^{-1}$, with an average of 0.0087 ppb m$^{-1}$.

The average gradient is approximately 20 times smaller than the afternoon gradient observed in the canopy (0.18 ppb m$^{-1}$ shown in Fig. 6). The positive ozone gradient (i.e., increasing ozone mixing ratio with increasing height) suggests ozone loss at the surface with continued atmospheric mixing. Comparing the three days of ozone mixing ratios to the wind speeds (not shown), lower wind speeds are associated with higher ozone mixing ratios over these 3 days, suggesting production and accumulation in the forest. When the wind speed is higher than 3 m s$^{-1}$, lower ozone levels are observed suggesting that the increase of wind speed dilutes ozone and decreases the ozone mixing ratio, or there may be insufficient time for ozone production chemistry to occur. Higher wind speeds should also be associated with stronger turbulence, due to enhanced wind shear. The stronger turbulence and mixing could lead to a weaker gradient; however, no correlation between the gradient and the wind speed is seen here.

The nearest regular ozone sonde launches are from Stony Plain, more than 400 km SSW of the YAJP site; however, these launches are rarely done in the afternoon. Between 1986 and 2008, there were 17 launches in the month of June at approximately 17:00 local time (https://woudc.org/). The ozone gradients measured by these sondes between heights of 50 m and 300 m ranged from −0.0014 to 0.0307 ppb m$^{-1}$, with an average of 0.0081 ppb m$^{-1}$. This average is approximately 7% smaller than our measured gradient average (0.0087 ppb m$^{-1}$) over the same height range. By comparison, for the month of July there were 21 launches with an average of 0.0103 ppb m$^{-1}$, approximately 20% greater than our measured gradient average.

420 Hence, these measurements demonstrate the substantial variability of the short-term, in-canopy vertical profiles, which are affected by shading. The longer-term measurements show a clear diurnal variation in the gradient, with a weaker gradient in the morning and stronger gradients overnight. Above-canopy gradients in the afternoon show less variability and are relatively consistent with gradients derived from ozone sonde measurements outside the region.

### 3.3 Above-canopy Ozone Mixing Ratio Comparison

425 The modeled ozone mixing ratio for each configuration listed in Table 1 is compared to measured values (both at heights of 22 m) in Figure 8. Statistics (ratio of modeled to observed averages, RMS error, and $R^2$) for the runs are listed in Table 1. Running the model with a constant NO value (configuration #1) results in an average modeled ozone value equal to the average measured value (Ratio of 1.0) as well as the lowest RMS error (10.7 ppb) of all the configurations. We refer to this configuration as the base case. As discussed in Section 2.4, initial test runs demonstrated miss-alignment of plumes when 430 using the GEM-MACH NO values as input to the 1D canopy model. This is demonstrated by the results of configuration #2, with a relatively high RMS (15.3 ppb) and low $R^2$ (0.352). Using NO as a function of wind direction (based on GEM-MACH output, but with values corrected according to wind direction) improves the results relative to the GEM-MACH output with an improved RMS (13.8 ppb) and the highest $R^2$ (0.499). Although using NO as a function of wind direction (#3) improves the correlation relative to the base case (with a constant NO), the model underpredicts the ozone (ratio of 435 0.88) and the RMS is higher when the NO varies with wind direction (13.8 versus 10.7 ppb). As demonstrated in Figure 8, using either the GEM-MACH NO or the NO based on wind direction results in complete removal of ozone on most nights, which is not supported by the observations.

Including increased NO near the surface below a height of 3 m (configs. #4 and #5) improves the model performance relative to the base case, with slightly lower ratios (0.98 and 0.91 versus 1.0) and higher $R^2$ values (0.438 and 0.474 versus 440 0.423). Very little difference is apparent between the base case and these two configurations in Figure 8; however, modifications of the NO profile are expected to have a more significant effect on the modeled ozone gradients, discussed in the following section.

Removing dry deposition from the model (config. #6) increases the measurement/model ratio (1.20), increases the RMS error (12.2 ppb), and reduces the $R^2$ (0.343), while doubling the deposition rate to 0.8 cm s$^{-1}$ (config #8) reduces the ratio 445 (0.91), and increases the $R^2$ value (0.449) and results in the same RMS error as the base case (10.7).

Although turbulent mixing is constrained above the canopy-top by measurements of turbulent kinetic energy ($e$) and the parameterization of Eq. 1, we also explore the effect of varying the $K$ values by two orders-of-magnitude (especially given that this is anticipated to affect the modeled ozone gradient). For these two configurations, the $K$ values are modified (at all heights) by a factor of 0.5 (config. #9) and a factor of 2 (config. #10), which we designate as weaker (#9) and stronger (#10) 450 mixing, respectively. The weaker mixing results in an improved RMS error (10.2 ppb) relative to the base case (10.7), while stronger mixing increases the RMS error (11.6 ppb). Weakening the mixing results in a model underestimation of ozone (ratio of 0.90) and strengthening the mixing results in a model overprediction (ratio of 1.09). Hence, the results imply that

elevated surface NO (accounting for surface NO emissions), stronger ozone deposition, and weaker mixing may give better model performance, but different results are demonstrated by different statistical measures. In the next section, output from these model configurations is compared to measured gradients within and above the canopy.

Except for configurations #2 and 3, most of the model runs follow a similar diurnal pattern (Fig. 8) which reproduces the daily cycle of ozone. However, there are differences between the measured and modeled ozone, most prominently in the mornings of June 22 and 25 and through the day of June 26. The worst aspect of the model behaviour seems to be an underprediction of ozone in the mornings between 06:00 and 12:00, except for the 26th where ozone is overestimated throughout the day and evening (12:00 to 00:00). The 26th was a cloudy day with peak PAR near 900 μmol m$^{-2}$ s$^{-1}$ (compared to 1300 to 1600 μmol m$^{-2}$ s$^{-1}$ for the other days), but this is not translated into lower ozone in the model. The difference on the 26th could be due to elevated NO at the measurement site that is not included in the model due to the assumption of constant NO. The GEM-MACH model (results not shown here) gives elevated ozone and low NO on the 24th, and lower ozone and higher NO on the 26th, which is consistent with this explanation. Observed winds are predominantly from the south on both the 24th and 26th, but wind speeds on the 26th are lower and wind direction is more variable, which could lead to recirculation of emitted pollutants back to the tower location.

**3.4 Above- and In-Canopy Vertical Gradient Comparison**

Modeled vertical ozone profiles are compared to the measured tethersonde profiles in Figure 7. These hourly profiles (shown for 4 hours at the same time of day as the measurements) are averages for the 6 days of the model run. The modeled vertical profiles demonstrate nearly linear vertical gradients in the 20 to 300 m range. This range is used to calculate the 2-point gradient $d[O_3]/dz$) shown in Figure 9. Figure 9 compares the modeled gradients for all model configurations by hour-of-day (averaged for the 6 model days) with the tethersonde measurements (Fig. 9a) and the long-term gradients within the canopy (Fig. 9b). The tethersonde measurements were made between ~11:00 and ~18:00 and range from −0.0014 to 0.02 ppb m$^{-1}$, with an average of 0.0087 ppb m$^{-1}$. In this time range (11:00 to 18:00) the modeled gradients above the canopy range from near zero (with either no deposition or stronger mixing, configs. #6 and 10) to 0.024 ppb m$^{-1}$ (with weaker mixing, config. #9). The weaker mixing configuration (#9) gives model results which are closest to the measured values. Both the increased surface NO and the strong depositions configurations (#5 and #8) produce similar diurnal variation of the above-canopy gradients. In the afternoon, these model gradient values are within one standard deviation of 4 of the 8 measured gradient values (6 observed gradients are higher than the modeled values and 2 are lower). The average modeled gradient value for this configuration (0.0066 ppb m$^{-1}$) is 76% of the average measured value, which can be considered good agreement given the amount of variability in the observations.

Modeled vertical profiles within the canopy are compared to the measured vertical profiles in Figure 5. Stronger curvature is seen in these profiles in the upper half of the canopy (see also the below-canopy profiles in Figure 7). To compare the model profiles to the long-term measurements, the gradient between 2 m and 25 m is calculated from the model output. Figure 9b demonstrates the average and median hourly gradient between heights of 2 m and 25 m for the 10 model configurations. This

can be compared directly to average in-canopy gradient measurements made 4 times daily between 27 March and 22 June 2019. The measured overnight gradient is near 0.35 ppb m$^{-1}$, while the measured morning gradient is near 0.18 ppb m$^{-1}$. Although the relative cold in later March and the potential presence of snow might affect the gradients, recalculating the gradients excluding 27 March to 30 April results in an average difference ~1% relative to the complete period. The model was run for a different period (21 – 27 June 2018), since this is when continuous ozone measurements at the canopy top were available. The average air temperature in the 27 March to 22 June 2019 period was 8.3°C compared to 21.0°C in the 21 – 27 June 2018 period, while the average PAR was similar for the two periods (393 and 401 µmol m$^{-2}$ s$^{-1}$ respectively). Assuming the two time periods are comparable, the model configurations tend to underestimate the overnight gradients, while estimations of the afternoon gradient is underestimated or overestimated depending on the configuration. All of the model configurations, with the exception of no deposition (#6), demonstrate a diurnal variation which is opposite to the measured pattern. The measurements demonstrate a stronger gradient overnight (as is also seen in the diurnal ozone gradient measurements of Wu et al., 2016 in a mixed temperate forest), while the model generally predicts smaller gradients overnight and higher gradients in the afternoon for nearly all configurations. The only model configurations which are consistently within one standard deviation of the measurements are the the base case (#1), weak deposition configuration (#7) with $v_d = 0.2$ cm s$^{-1}$ and the strong mixing (#10), which all show good agreement in the morning, while the base case (#1) shows good agreement in evening and the other cases (#7 and #10) show good agreement in the afternoon. The weaker mixing configuration (#9) shows the best agreement with the overnight gradients but overestimates the afternoon and evening gradients.

Rannik et al. (2012) demonstrate a strong diurnal cycle of deposition velocity, averaging 0.2 cm s$^{-1}$ at night compared to more than 0.5 cm s$^{-1}$ during the day (in the summer months). Figure 9b demonstrates that a lower deposition velocity results in a smaller gradient (compare #6, 7, and 8). Although our model unrealistically assumes a deposition velocity that is constant with time, the results suggest that modeling a deposition velocity that is lower at night and a higher during the day would result in a weaker gradient at night and a stronger gradient during the day (relative to a constant value). This would further increase the difference between the model results and the observations, which show stronger gradients at night.

Hence, the combined results suggest three possible model corrections: 1) ozone deposition could be weaker in this forest relative to previous studies such as Rannik et al. (2012) and Wu et al. (2016); 2) in-canopy mixing may be overestimated during the night; and 3) in-canopy mixing may be underestimated during the day. We ran various combinations of these 3 modifications (deposition velocity of 0.2 cm s$^{-1}$, decreased mixing before noon, and increased mixing after noon) and found that using $v_d = 0.2$ cm s$^{-1}$ (#7) combined with a decrease in $K$ by a factor of two between 0:00 and 12:00 (#9 at night only) reproduced the observed pattern of strong gradients overnight and weak gradients through the day (labelled "#7 with #9" in Fig. 9b), however, the evening gradient is still underpredicted by this combined configuration. All other combinations of the 3 modifications (not shown in Fig. 9b) gave results very similar to other previously discussed configurations and did not reproduce the observed diurnal pattern of the gradients. Hence, these model results suggest that mixing is overestimated overnight, and that deposition velocity is in the range of 0.2 to 0.4 cm s$^{-1}$. Since there is discrepancy in the model-

measurement comparison in the afternoon compared to the evening, no single model configuration (or combination) reproduces both the afternoon and evening gradients successfully and it is not possible to more accurately infer the deposition velocity with this model. The deposition velocity may be lower relative to the Rannik et al. (2012) study ($v_d =$ 0.2 cm s$^{-1}$ compared to 0.4 cm s$^{-1}$) due to the lower density of the forest. The LAI of the boreal forest in Rannik et al. ranged from 6 to 8, while the LAI of the mixed temperate forest described in Wu et al. was 4.6, both much higher than the LAI of 1.17 at this site. The estimated deposition may also be affected by the "big leaf" assumption. Spreading the ozone deposition through the vertical profile of the canopy would likely reduce the gradient relative to deposition at the surface only. Hence, the "big leaf" assumption may lead to an overestimation of the deposition rate, further increasing the difference between the model results and the Rannik et al. study. The improvement of the model-measurement comparison at night by reducing the eddy diffusivity suggests that Equations 2 and 3 may underestimate the stability at night when fluxes ($u_*$ and $\overline{w'T'}$) are weaker.

Although, this combined configuration (#7 with #9) is not listed in Table 1 or shown in Fig. 8, the O$_3$ mixing ratio at a height 22 m with this combined configuration of is similar to that of configuration #7 (weak ozone deposition). The average ratio (modeled to measured ozone) for the combined configuration is 1.05, the RMS error is 10.9 ppb, and the $R^2$ is 0.402 (compared to 1.07, 11.0 ppb, $R^2 = 0.396$ for config. #7 or 1.00, 10.7 ppb, $R^2 = 0.423$ for the base case #1). Hence, modeling weak deposition with the inclusion of weaker mixing overnight gives a better model performance (for ozone mixing ratio above the canopy) relative to the modeling weak deposition with unmodified mixing; but the performance is slightly worse than the base case with moderate (0.4 cm s$^{-1}$) deposition and unmodified mixing. However, neither the base case (#1) nor the weak deposition alone (#7) can fully reproduce the observed diurnal variation of the gradients.

## 4 Conclusions

Although there is variability in the results, ozone measurements segmented by wind direction generally support the results of Cho et al. (2017) and Aggarwal et al. (2018), which demonstrate no significant increase in ozone levels (or some ozone reduction) when winds are from the direction of oil sands production relative to background (forested) sources.

In-canopy ozone measurements at heights of 2 m and 25 m indicate a stronger gradient in the evening and overnight (near 0.35 ppb m$^{-1}$) with a weaker gradient in the morning (near 0.18 ppb m$^{-1}$) and the afternoon (near 0.24 cm s$^{-1}$). This is consistent with increased mixing within the canopy in the afternoon (see our companion paper, Jiang et al., 2022) and with the ozone gradient measured in a mixed temperate forest (Wu et al. 2016). Model analysis can reproduce the in-canopy gradients overnight when the diffusivity coefficients are reduced by a factor of 2 overnight. The afternoon gradient is reproduced with a modeled ozone deposition velocity of $v_d = 0.2$ cm s$^{-1}$, while the evening gradient is reproduced with a modeled ozone deposition velocity of $v_d = 0.4$ cm s$^{-1}$. Hence, we infer that the ozone dry deposition velocity for this forest location is between 0.2 cm s$^{-1}$ and 0.4 cm s$^{-1}$.

Model simulations were shown to be most sensitive to the assumed input NO concentration and the coefficient of vertical diffusivity; height-dependent concurrent observations of $e$, NO, and $O_3$ are recommended for future work. Ideally, fast ozone (and NO) analyzers could directly measure fluxes (as in Finco et al., 2018) to directly determine deposition velocity and to compare these values to gradient measurements and dry deposition parameterizations.

The reduced overnight mixing may suggest that modeling nighttime stability using the Obukhov length (Eq. 2) does not account for the increased stability within a canopy associated with canopy decoupling, which further demonstrates a known weakness in using a local gradient-diffusion model ($K$-theory) to model nighttime canopy mixing (e.g., Lee and Mahrt, 2005). Moderate overnight canopy decoupling was measured at this site as described in Jiang et al. (2022). Further investigation is needed to improve how this decoupling is incorporated into the 1D canopy model. We note that the

parameterization of in-canopy turbulence employed here (M17) imposes a generic shape of observed turbulence profiles below the canopy starting from an above-canopy $K$ value, to attempt to capture this decoupling. However, the local turbulence profile may differ from the generic profile of M17; additional observations of turbulent kinetic energy ($e$) with height would assist in generating a location-specific $K$ profile for modelling.

Although the deposition velocity resulting in the best fit to observed $O_3$ profiles is lower here than deposition velocities

reported in boreal forests such as Rannik et al. (2012), this could be due to the sparser forest and lower LAI at this location relative to those locations. This lower deposition velocity for ozone contrasts with the higher deposition velocity for $SO_2$ for this region as described in both Gordon et al. (2022) and Hayden et al. (2022), suggesting that the increased $SO_2$ dry deposition could be related to a chemical process such as surface acidity that does not affect ozone deposition. There is also uncertainty associated with the location of the "big leaf" at the forest floor. In future work, the model could be developed to

investigate the effect of a vertical distribution of uptake throughout the canopy height.

### Code/Data Availability

The model code and the field data described in this study will be posted in a data repository prior to publication.

### Author Contributions

XZ and MG wrote the original draft of this work and performed the analysis. MG and PM conceptualized the field study.

PM acquired funding. XZ, TJ, and MG performed the field experiments. JD provided sonde instrument and instruction tests. DT provided Stony Plain ozonesonde data. XZ, MG, PM, TJ, JD, and DT reviewed and edited the work.

### Competing Interests

The contact author has declared that none of the authors has any competing interests.

## Acknowledgements

Funding to XZ, MG, and TJ provided by Environment and Climate Change Canada (GCXE20S044). We acknowledge the shared data, technical support, and assistance of the Wood Buffalo Environment Association (WBEA) of Alberta. This research was enabled in part by support provided by the Digital Research Alliance of Canada (alliancecan.ca).

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

**Table 1. Model configurations. NO input and deposition velocity ($v_d$) are given for each model configuration. When a single NO mixing ratio is given, it is input as constant (in height and time). When two NO mixing ratios are given, the first is for height less than 3 m and the second if for heights above 3 m (both constant in time). Resulting statistics for measured and modeled ozone at a height of 22 m for the 6-day model runs show: ratio of averages (modeled to observed), RMS error, and coefficient of correlation ($R^2$). The best value in each category is underlined.**


| Config. # | NO [ppb] | $v_d$ [cm s$^{-1}$] | Description | Ratio | RMS [ppb] | $R^2$ |
|---|---|---|---|---|---|---|
| 1 (base) | 0.05 | 0.4 | Constant NO (with time and height) | 1.00 | 10.7 | 0.423 |
| 2 | Variable | 0.4 | NO from GEM-MACH | 1.21 | 15.3 | 0.352 |
| 3 | Variable | 0.4 | NO as a function of Wind Dir. | 0.88 | 13.8 | 0.499 |
| 4 | 0.05, 1 | 0.4 | Elevated surface NO (1 ppb) | 0.98 | 10.7 | 0.438 |
| 5 | 0.05, 5 | 0.4 | Elevated surface NO (5 ppb) | 0.91 | 10.8 | 0.474 |
| 6 | 0.05 | 0 | No ozone deposition | 1.20 | 12.2 | 0.343 |
| 7 | 0.05 | 0.2 | Weak ozone deposition | 1.07 | 11.0 | 0.396 |
| 8 | 0.05 | 0.8 | Strong ozone deposition | 0.91 | 10.7 | 0.449 |
| 9 | 0.05 | 0.4 | Weaker mixing ($0.5K$) | 0.90 | 10.2 | 0.420 |
| 10 | 0.05 | 0.4 | Stronger mixing ($2K$) | 1.09 | 11.6 | 0.417 |


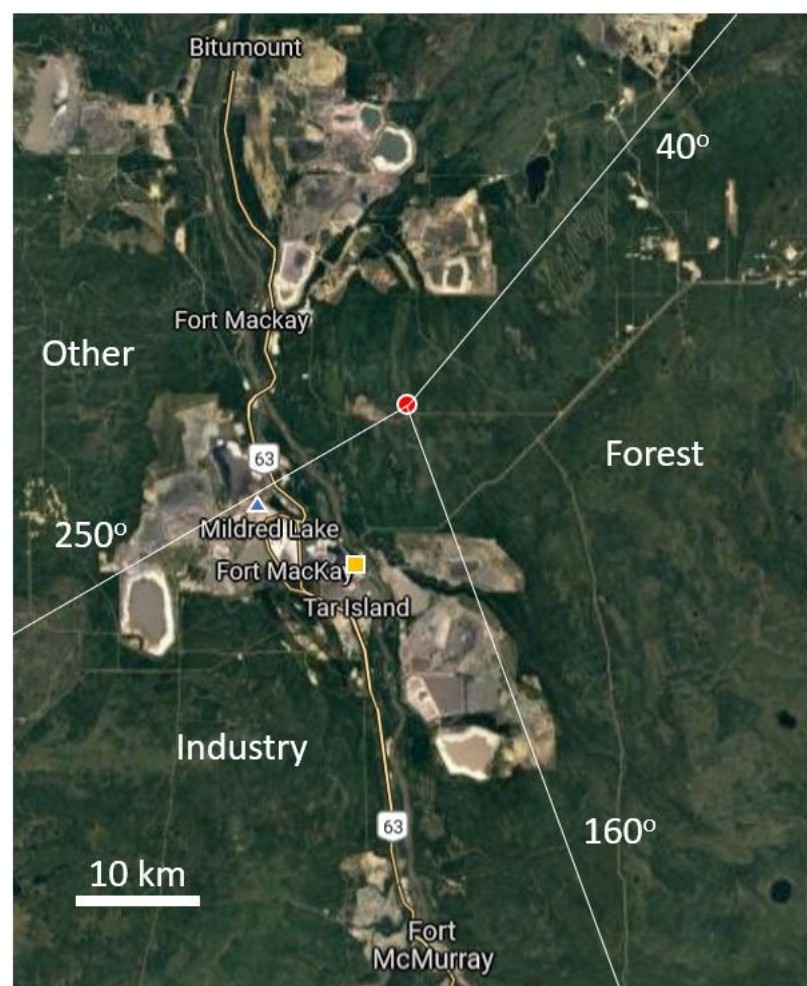

**Figure 1: The study area and surrounding areas of the measurement sites showing the YAJP tower location (red dot), Syncrude (blue triangle) and Suncor (yellow square) stack locations, and wind sectors (industry, forest, and other) used in the following analysis (based on $SO_2$, aerosol, and $CO_2$ measurements). Delineation of the wind sectors is shown by white lines with corresponding wind directions. Map image is © Google Maps.**


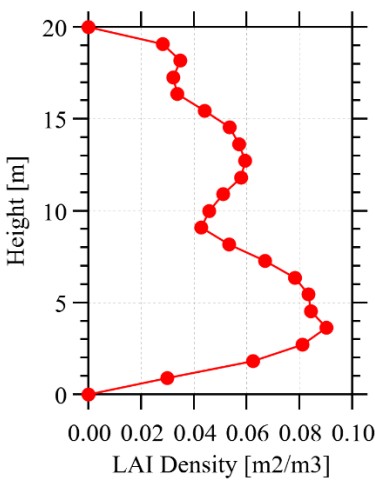

**Figure 2: The LAI profile near the YAJP tower.**

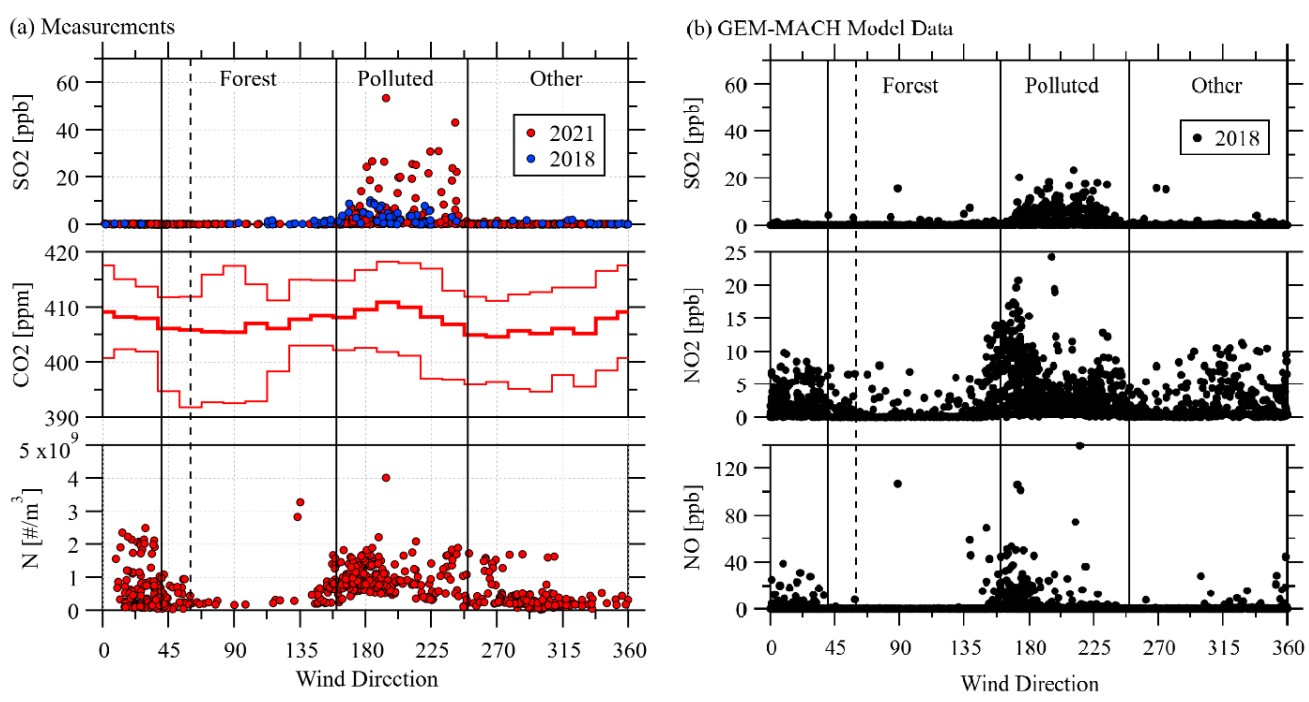


**Figure 3: (a) Measured SO₂, CO₂, and aerosol number concentration (*N*) with wind direction (ºN). SO₂ measurement from 9-19 Jun 2018 are shown as blue dots. SO₂ and aerosol measurements from 7-25 Aug 2021 are shown as red dots. SO₂ data and aerosol data are reproduced here from the companion papers Gordon et al. (2022) and Jiang et al. (2022). CO₂ mixing ratios were measured from 2017 to 2021 and are shown as median, 25th, and 75th percentiles in 15º bins. (b) GEM-MACH model output of**

**SO₂, NO, and NO₂ for comparison modeled from 1 Jun to 17 Aug 2018. Vertical lines delineate the 3 sectors used in the analysis (forest, polluted, other) also shown in Fig. 1. The dashed line shows the generator sector (40 – 60º). Data in the generator sector are discarded if the generator was used during that period.**

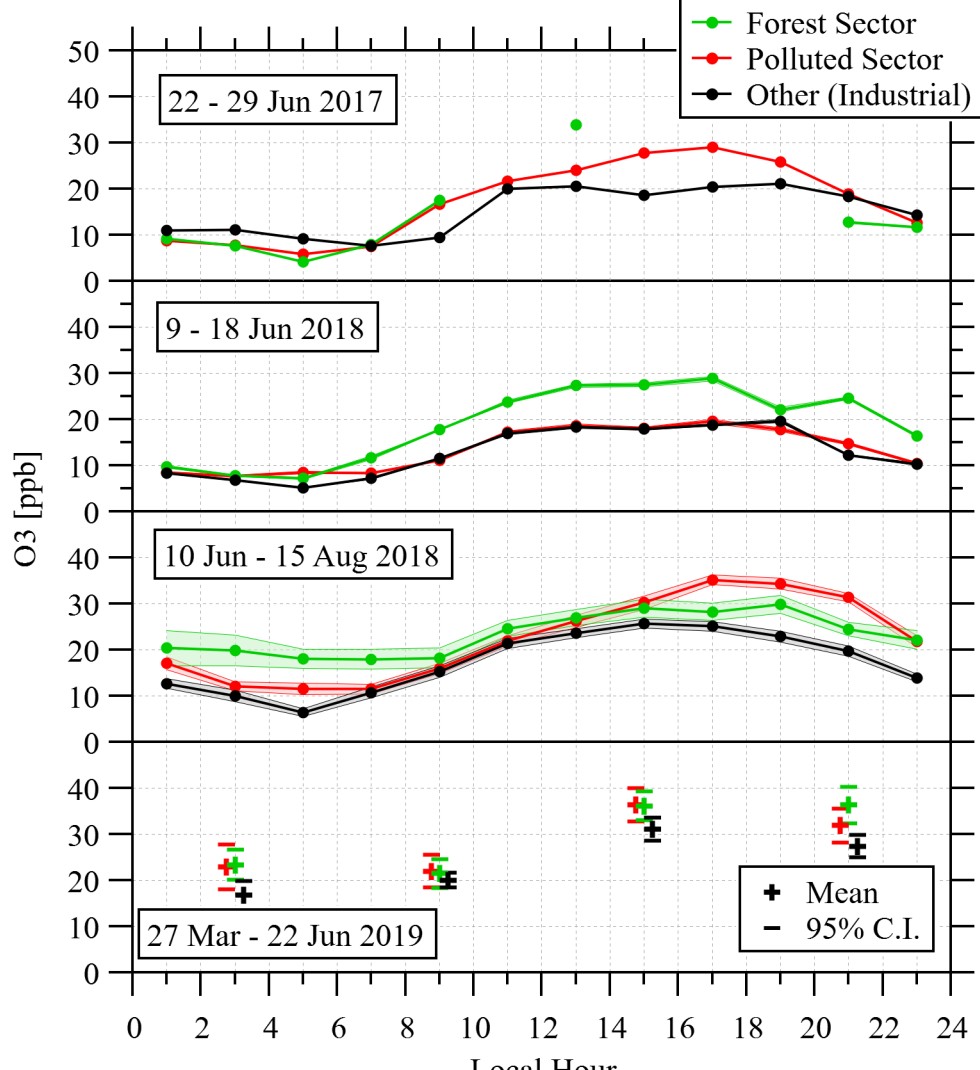

**Figure 4: Diurnal cycle of O₃ (truncated mean values in 2-hour bins) from three defined sectors for four measurement periods. Shading shows the 95% confidence interval (shown by error bars for the 2019 data). In 2019, the instruments were run from solar power and were activated only 4 times per day to conserve power. The 2019 sectors are offset at each of the 4 times for ease of comparison, although all measurements were made at the same time. Sectors are defined in the text and are shown in Figs. 1 and 3. Measurement heights were approximately 16 m during 2017, 25 m for 9 - 18 June 2018, 19 m for 10 Jun - 15 Aug 2018, and 25 m for 2019 (the canopy height is approximately 19 m).**


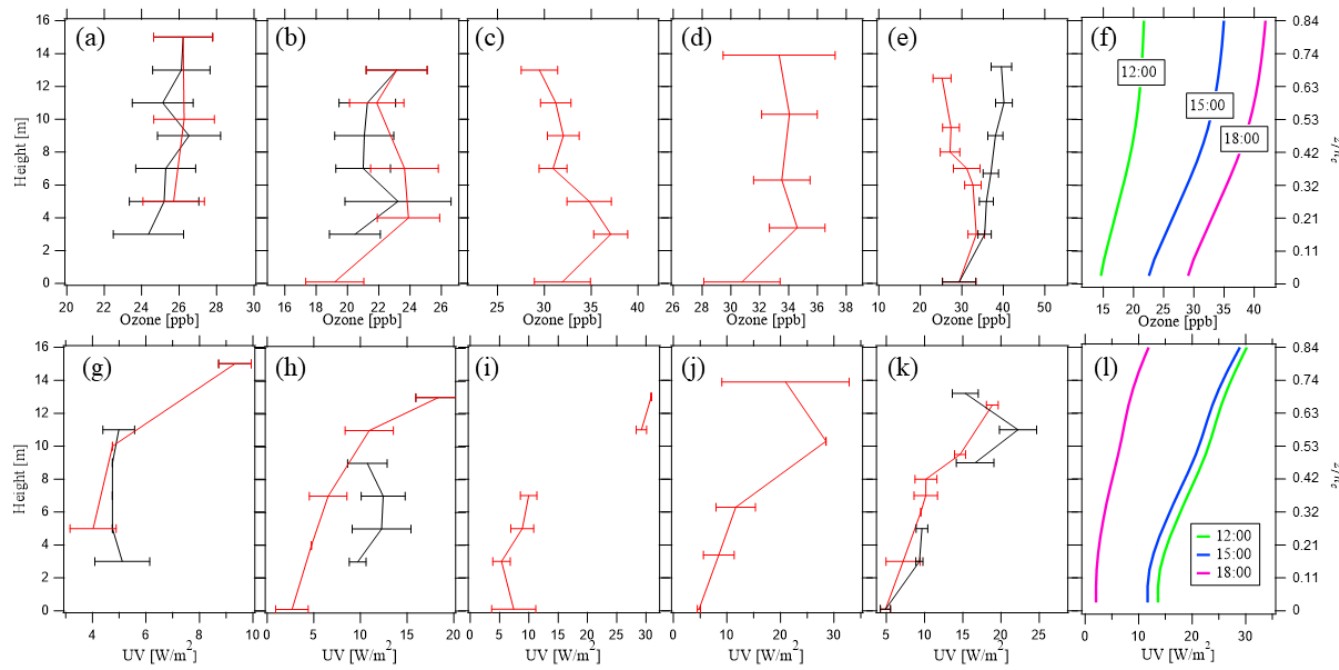


**Figure 5: Vertical profiles of ozone (a-f) and UV (g-l) from the tower pulley system within the canopy. Right axis shows heights relative to the canopy height ($h_c$ = 19 m). Measurements are shown in (a-e) and (g-k) and model output (for 3 sample hours averaged over the model run) is shown in (f) and (l). Measurement dates (all 2017) and approximate times are: 18:00, 22 Jul (a,f); 17:00, 24 Jul (b,g); 15:00, 25 Jul (c,h); 15:00 to 18:00, 26 Jul (d,i); 12:00, 27 Jul (e,j). Black lines show ascending measurements,**
**and red lines show descending measurements. Error bars show standard deviations at each height within each 15-min measurement period.**

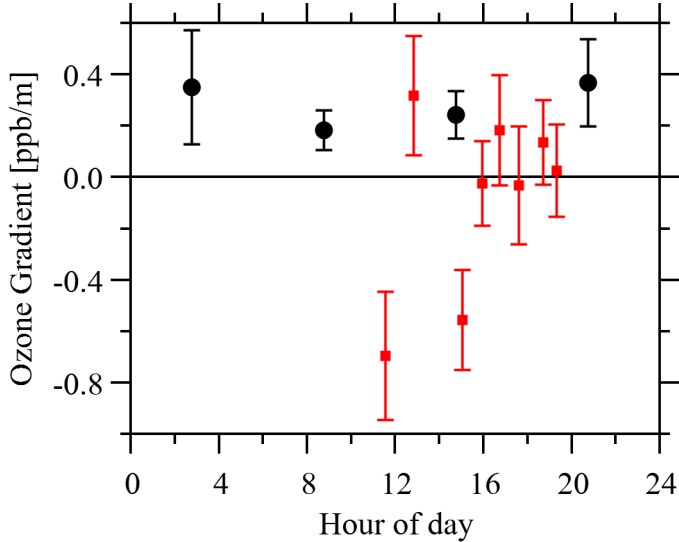

**Figure 6: Within-canopy ozone gradients as $dC/dz = (C(z_u) - C(z_l))/(z_u - z_l)$. Black circles are gradients between heights of $z_u = 25$ m and $z_l = 2$ m for the period 27 Mar to 23 Jun 2019. Red squares are the average gradients from the 2017 in-canopy profiles shown in Fig. 5 (22 to 27 Jul 2017) with upper height ranging from $z_u = 12.5$ to 15 m and lower height ranging from $z_l = 3$ to 5 m. Error bars show one standard deviation.**

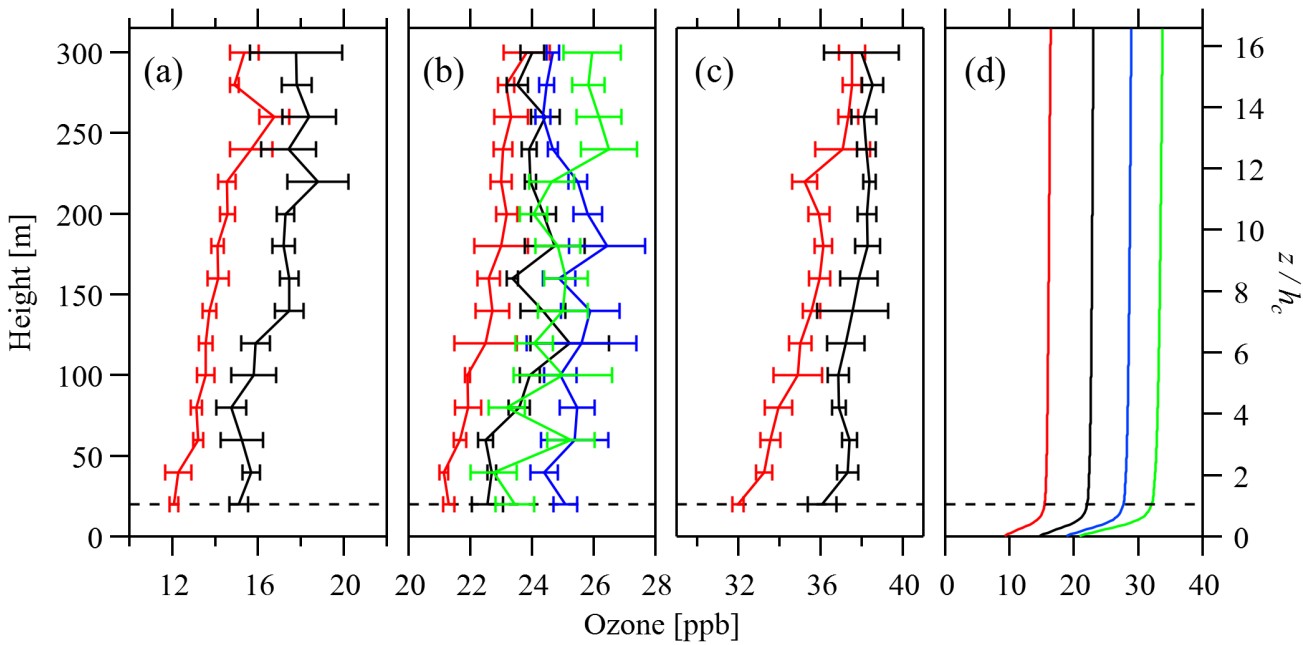

**Figure 7: Ozonesonde profiles from above the canopy. Right axis shows heights relative to the canopy height ($h_c$). Measurement dates (all 2018) and times are (a) 13 Jun 10:45-11:30 (red), 12:30-13:20 (black); (b) 15 Jun 12:00-12:30 (red), 12:40-12:55 (black), 13:05-13:15 (blue), 13:55-14:15 (green); (c) 16 Jun 17:00-17:30 (red), and 17:50-18:00 (black). Model output (d) is averaged over the model run for 11:00 (red), 12:00 (black), 13:00 (blue), and 14:400 (green). Error bars show standard deviation in each 20-m height interval. Dashed line shows canopy height ($h_c = 19$ m).**

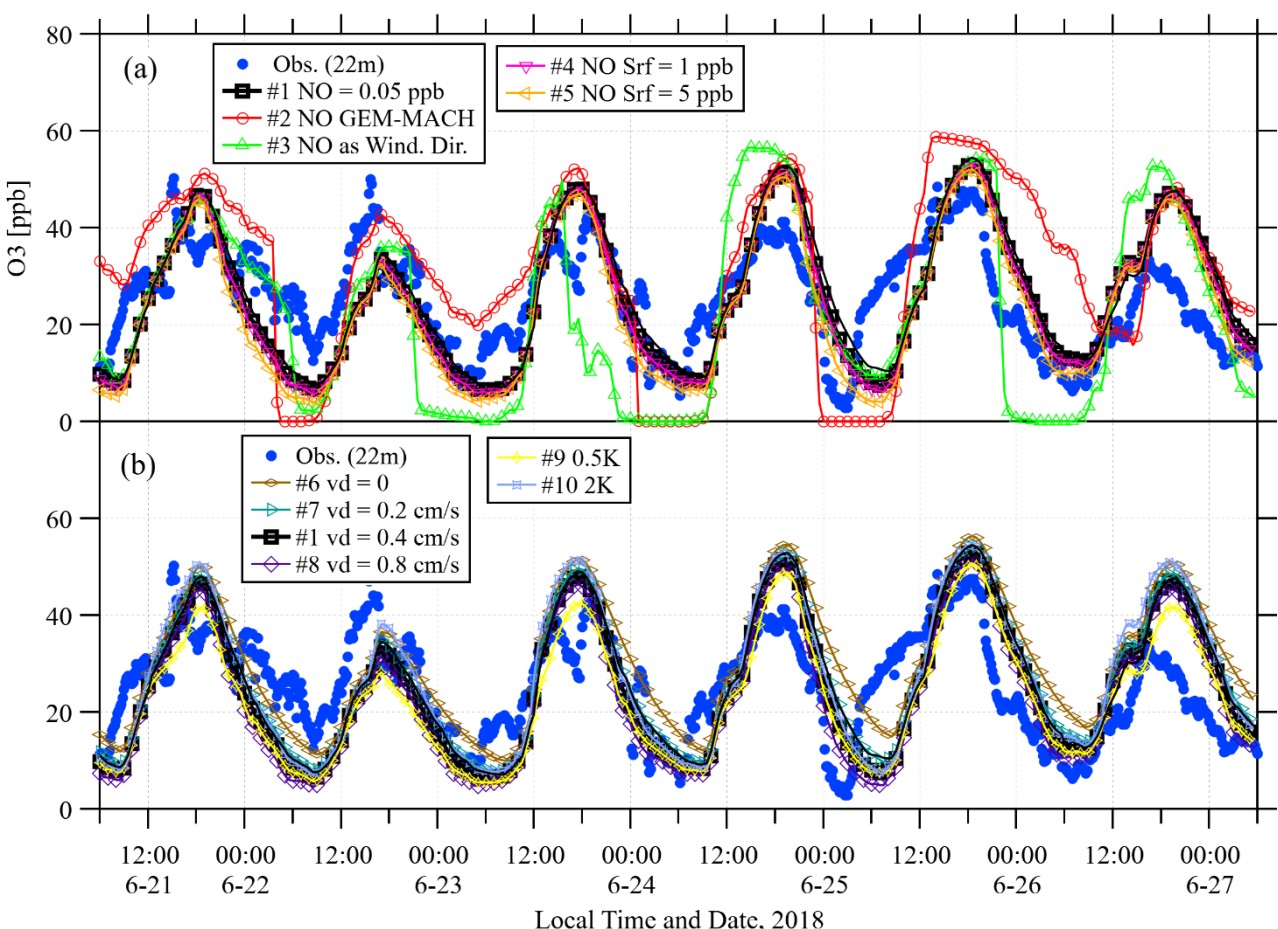

**Figure 8: The ozone measurements and model output (at a height of 22 m) for the 10 configurations listed in Table 1. Observations and model output for configurations which modify NO are shown in (a). Observations and model output for configuration which modify deposition velocity and eddy diffusivity are shown in (b). The first 12 hours (not shown) are excluded from the results as model spin-up. Observation frequency is every 10-minues. Model output is 30-min, but only every second marker is shown for clarity.**

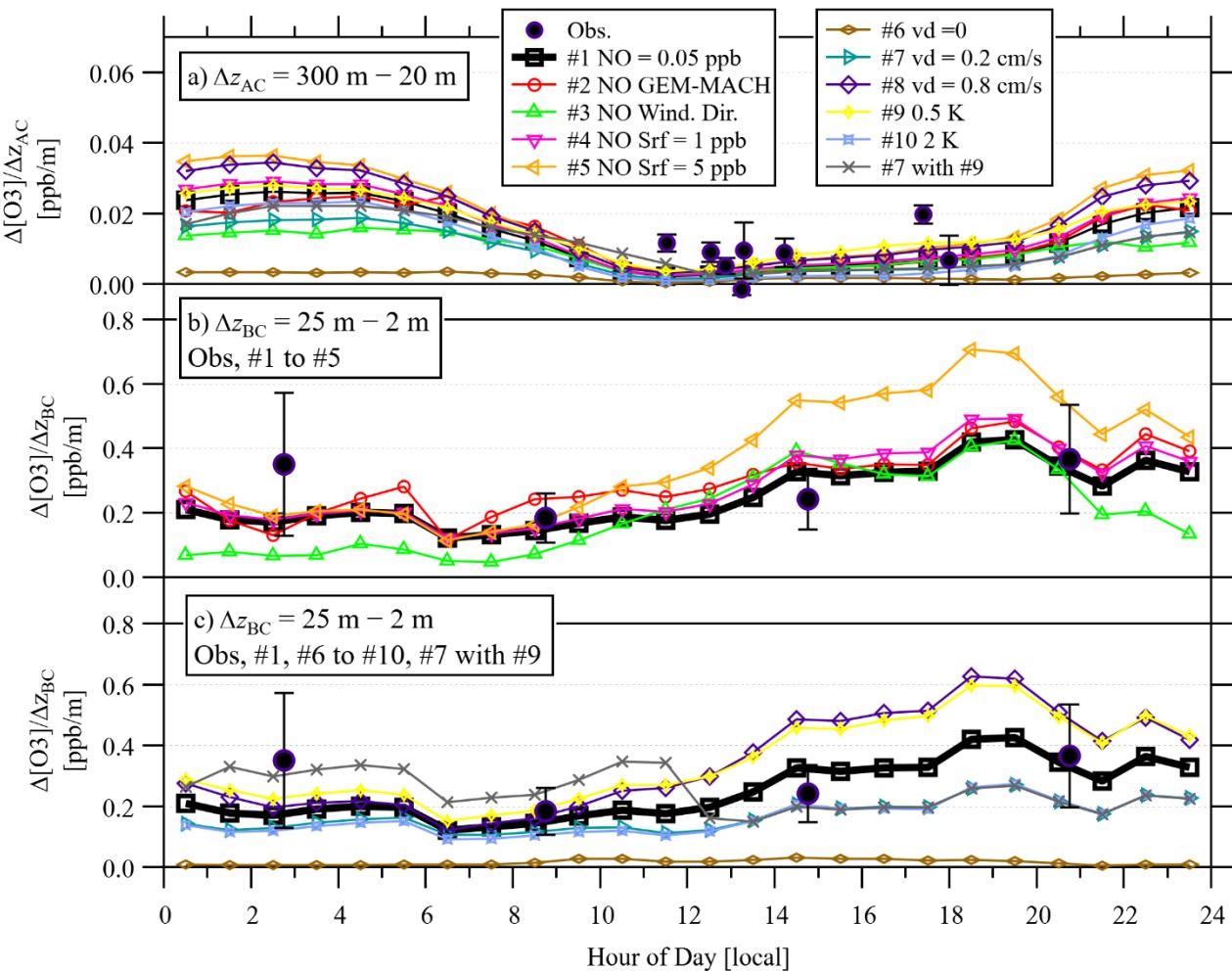

**Figure 9: Hourly gradients from the 10 model runs as listed in Table 1. The average ozone gradient is shown (a) above the canopy (AC) between heights of 20 and 300 m and (b & c) below the canopy (BC) between heights of 2 and 25 m. The black solid circles show measured gradients from the tethersonde above the canopy (Fig. 7) and the long-term below-canopy gradients (Fig. 6). For clarity, model output for configurations which modify NO are shown in (b) and model output for configuration which modify deposition velocity and eddy diffusivity are shown in (c). Error bars show standard deviations of the measurements.**