# Peer review of "Ozone in the boreal forest in the Alberta oil sands region"

_Atmospheric Chemistry and Physics, 2023_

## Referee Comment (RC2)

**Ozone in the boreal forest in the Alberta oil sands region**

Xuanyi Zhang[1], Mark Gordon[1], Paul A. Makar[2], Timothy Jiang[1*], Jonathan Davies[2], David Tarasick[2]

[1]Earth and Space Science, York University, Toronto, M3J 1P3, Canada
[2]Air Quality Research Department, Environment and Climate Change Canada, Toronto, M3H 5T4, Canada
5 [*]Now at School of Environmental Studies, Guelph University, Guelph, N1G 2W1, Canada

*Correspondence to*: Mark Gordon (mgordon@yorku.ca)

**Abstract.** Measurements of ozone were made using an instrumented tower and a tethersonde located in a forested region surrounded by oil sands production facilities in the Athabasca Oil Sands Region (AOSR). Our observations and modelling show that the concentration of ozone was modified by vertical mixing, photochemical reactions, and surface deposition.
10 Measurements on the tower demonstrated that when winds are from the direction of anthropogenic emissions from oil sand extraction and processing facilities, the ozone mixing ratio in the forest is as much as 10 ppb lower than when winds are from the direction of undisturbed forest. This finding is supported by previous studies which suggest that surplus $NO_x$ from oil sands emissions results in ambient ozone titration. Gradients of ozone mixing ratio with height were observed using instruments on a tethered balloon (up to a height of 300 m) as well as a pulley system and 2-point gradients within the
15 canopy. Strong gradients (ozone increasing with height between 0.2 and 0.4 ppb m$^{-1}$) were measured in the canopy overnight, while daytime gradients were weaker and highly variable. A 1D canopy model was used to simulate the afternoon in-canopy gradient with reduced mixing overnight (suggesting high stability within the canopy), and an ozone deposition velocity of 0.2 cm s$^{-1}$. Sensitivity simulations using the model suggest the local NO concentration profile and coefficients of vertical diffusivity have a significant influence on the $O_3$ concentrations and profile in the region.

**1 Introduction**

20 Canada's largest oil sands deposits areas are found in the Athabasca Oil Sands Region (AOSR) of northern Alberta. The increasing oil sands production has led to increased environmental concern for the nearby forest ecosystem (Li et al., 2017). The processes  include surface mining to turn surface oil sands into crude oil, well injection to pump deeper bitumen onto the surface, extraction of bitumen from oil sands with the water-based process and the upgrading of
25 bitumen into hydrocarbon streams (Natural Resources Canada, 2016). Our study analyses how pollutant emissions associated with oil sands extraction modify ozone concentration and deposition in the surrounding forest. A boreal forest site was chosen, surrounded by oil sands processing facilities including those operated by Syncrude, Suncor, Canadian Natural Resources Limited (CNRL) as well as other facilities.

Ozone is a photochemical pollutant in the troposphere. It is produced there by the photochemical oxidation of carbon
30 monoxide, methane, and non-methane volatile organic compounds in the presence of nitric oxide (NO) and nitrogen dioxide
* * *
**Number: 1**      Author:    Subject: Sticky Note      Date: 5/8/2023 9:51:15 PM

Better wording would be:  ...that ozone is destroyed by reaction with nitric oxide emitted from oil and gas extraction operations (as well as NO resulting from photolosis of nitrogen dioxide). emissions),
* * *
**Number: 2**      Author:    Subject: Sticky Note      Date: 5/8/2023 9:51:15 PM

Vertical gradients ....
* * *
**Number: 3**      Author:    Subject: Inserted Text      Date: 5/8/2023 10:57:12 PM

of petroleum hydrocarbon extraction from oil sands
* * *
**Number: 4**      Author:    Subject: Inserted Text      Date: 5/8/2023 9:51:15 PM

,

[revised manuscript text omitted]

**Page: 3**

| | | | |
|---|---|---|---|
| Number: 1 | Author: | Subject: Sticky Note | Date: 5/8/2023 9:51:15 PM |

Not clear.

| | | | |
|---|---|---|---|
| Number: 2 | Author: | Subject: Sticky Note | Date: 5/8/2023 9:51:15 PM |

Need more detailed quality control data for 2B ozone monitors.

heights of 2 m and 25 m for long-term monitoring. The monitors were activated 4 times per day (02:00, 08:00, 14:00, and 20:00 local) for one hour duration. The sampling frequency during these time periods was twice per minute. To allow the instrument to equilibrate after each cold start, the measurement for each period was taken as the median value of the last 15-

100    minutes of measurement in each hour. These monitors remained operational until late June 2019.

During August 2021, there was a third summer intensive study at the tower. No ozone measurements were made during this field study, but $SO_2$ measurements (43i, Thermo Scientific and AF22e, Envea) at ground level and a height of 30 m and size-resolved sub-micron aerosol measurements (UHSAS, DMT) helped to further identify wind sectors bringing polluted air to the site. The $SO_2$ and aerosols measurements are discussed in our companion papers Gordon et al. (2022) and Jiang et al.

105    (2022) respectively.

Permanent instruments on the YAJP tower (solar powered) used for the following analysis include sonic anemometers (ATI Inc.) at heights of 29 m and 5.5 m, photosynthetically active radiation (PAR, LI-190, Licor Inc) measured at heights of 29 m, 15.9 m, and 2 m, and a gas analyzer ($CO_2$/$H_2O$, LI-7500, Licor Inc) at a height of 29 m.
[Figure]

The Wood Buffalo Environmental Association (WBEA) operates a meteorological tower identified as "1004" which is

110    approximately 540 m south of the YAJP tower. The 1004 tower measures hourly values of: temperature, relative humidity (RH) and winds at heights of 2, 16, 21, and 29 m; PAR at heights of 2, 16, and 21 m; and atmospheric pressure (at 2 m).

**2.2 Ozone Modeling**

To model photochemical processes, diffusion, and deposition in the canopy, we use a one-dimensional (1D) canopy model created by Makar et al. (1999). The model has 1001 levels in the vertical direction, and each level has 1-m spacing. The

115    model uses 30-minute interval input data. It includes 268 chemical reactions associated with 79 output species. In the 1-D canopy model, the rate of change of each chemical species mixing ratio ($C$) at each model level is calculated due to their emissions ($E$), chemical reactions ($f$) and diffusion (Eq. 1) at each layer. In Equation 1, subscript m represents different chemical species, subscript n represents the vertical layer, and $K$ is the eddy diffusivity.

$$\frac{\partial C_{mn}}{\partial t} = E_{mn} + f_{mn} + \frac{\partial}{\partial z}\left(K(z_n)\frac{\partial C_{mn}}{\partial z}\right),$$            (1)

120    Further modifications were made to the model outlined in Stroud et al. (2005) and Gordon et al. (2014) to include sesquiterpenes, modify the diffusion code, and include surface deposition.

Model input variables are updated every 30 minutes. The model uses input data of pressure, PAR, and RH at a single height. Temperature and NO are input for the lowest 50 levels with 1 m spacing and the turbulent diffusion coefficient ($K$) is input for all 1001 levels at 1 m spacing. Temperature profiles were linearly interpolated using the 2, 16, 21, and 29 m

125    measurements from the 1004 tower. Initially, the temperature was set as constant above a height of 29 m, but test runs demonstrated improvement by assuming a dry adiabatic lapse rate (0.0098 K m$^{-1}$) above this height. As a sensitivity test, we run versions of the model with constant temperature above 29 m and with a dry adiabatic lapse rate above this height (Section 3.5). Pressure, PAR, and relative humidity are required as model inputs at the canopy height. These variables were

**Page: 4**

Number: 1        Author:    Subject: Sticky Note        Date: 5/8/2023 9:51:15 PM
Missing quality control and calibration data.

[revised manuscript text omitted]

Number: 1    Author:    Subject: Sticky Note    Date: 5/8/2023 9:51:15 PM
Wind sectors are not labeled in Figure 1.

Number: 2    Author:    Subject: Sticky Note    Date: 5/8/2023 9:51:15 PM
Probably none of the sector differences are statistically significant, but are well within the measurement uncertainty and variability of the data. So, this whole discussion about differences should be omitted.

[Figure]

although this is not the case through the afternoon/evening for the longer period 2018 (2 months) measurements, where ozone is lower from the *forest* compared to the *polluted* sector. In 2017 and the longer period (2 months) in 2018, the ozone levels when winds are from the *polluted* sector are higher than the *other* (primarily industrial but not forest) sector. The longer period measurements in 2018 (2 months) and 2019 (~3 months) demonstrate strongly elevated overnight ozone levels transported from the *forest* sector, likely representing background air unmodified by oil sands emissions. For these two periods, ozone transported from the *polluted* sector has a generally higher mixing ratio relative to the *other* (industrial) sector, possibly indicating increased titration with increasing transport time from the more distant sources of the *other* sector. These results are similar to previous studies that demonstrate no significant increase in ozone levels with increasing oil sands development (Cho et al., 2017) or ozone levels in the vicinity of oil sands production that are equal to or lower than the background levels (Aggarwal et al., 2018), at these distances from the sources. Although the results shown here are highly variable, there is a decrease in ozone when the air is polluted, generally on the order of 10 ppb.

**3.2 Ozone Vertical Profiles**

Measurements of ozone mixing ratio and UV radiation within the canopy are shown in Figure 5. These measurements were made on the tower pulley system in the summer 2017 campaign. The overall trend of ozone is increasing with height when the UV radiation was also increasing with height, which is the same as the Chen et al. (2018) analysis. This gradient can be explained by ozone deposition from stomatal uptake and/or chemical reactions such as near-surface $NO_x$ titration (Chen et al., 2018; Finco et al., 2018). In many profiles, there is a peak of ozone mixing ratio within the canopy near a height of 4 m. Finco et. al (2018) found that the ozone mixing ratio in the mid-level of the forest canopy is about 2.5% higher than the ozone mixing ratio above the canopy. Higher UV radiation measurements correlate with higher ozone mixing ratios, which demonstrates the relationship of UV radiation to the ozone vertical variation based on the ozone photochemical production reaction. The shading effect (demonstrated by the UV measurements) is in good agreement with the LAI profile (Fig. 2), where the lowest UV values coincide with the higher LAI value in the lower canopy (between 2 – 8 m), while the shading above 10 m is generally less pronounced.

Longer-term measurements (from 27 March to 23 June 2019) at fixed heights of 2 m and 25 m were used to determine the ozone gradient, as shown in Figure 6. Measurements indicate a stronger gradient overnight (near 0.4 ppb m$^{-1}$) with a weaker gradient (near 0.16 ppb m$^{-1}$) in the afternoon. A strong gradient would be expected during the night due to increased stability and a greater decoupling of the air above and below the canopy (M17). During the daytime, gradients may be affected by both the decreased photolysis rates within the shaded canopy and the increased turbulence due to afternoon convection. The presence of the canopy may reduce mixing relative to an open space. Continued deposition and/or titration of ozone through the night when there is little mixing will create a stronger gradient as ozone is removed within the canopy. Wu et al. (2016) demonstrate a similar diurnal trend of ozone gradient at a mixed temperate forest in summer with gradients near 0.35 ppb m$^{-1}$ overnight and 0.15 ppb m$^{-1}$ through the day. Our modeling efforts in the following section will attempt to replicate this diurnal variation.

Number: 1    Author:    Subject: Sticky Note    Date: 5/8/2023 9:51:15 PM
No statistical evaluation of the data.

Number: 2    Author:    Subject: Sticky Note    Date: 5/8/2023 9:51:15 PM
For radiation measurements perfect leveling of the light sensor is critical.  How was that accomplished on a moving pulley?

Number: 3    Author:    Subject: Sticky Note    Date: 5/8/2023 10:59:37 PM
Not a trend.  This is a vertical profile.

Number: 4    Author:    Subject: Sticky Note    Date: 5/8/2023 9:51:15 PM
Really not clear if the ozone loss is from reaction with NO or from stromatal uptake or deposition to surfaces and soil.

Number: 5    Author:    Subject: Sticky Note    Date: 5/8/2023 9:51:15 PM
Really not that obvious.  I don't see any obvious consistent ozone profile behavior that stands out of the variability and measurement uncertainty of the data.

Number: 6    Author:    Subject: Sticky Note    Date: 5/8/2023 11:00:51 PM
There is no plausible reason that ozone should be correlated with radiation as ozone production (dependent on radiation) would happen slower than the time scale of mixing.  Further, it's unlikely that there is significant ozone production in this low NOx environment that occurs on time scales that are shorter than the air transport within the domain.

[Figure]

The short-term summertime gradient measurements within the canopy (Fig. 5 and red squares in Fig. 6) are much more
290 variable than the other time periods. The gradients are likely sensitive to short-term variation in ozone mixing ratio during
the profiles due to changes in wind direction. These short-term gradients were determined using the difference between the
highest measurement and the measurement near a height of 3 m. The near surface measurements were not used for this
purpose because of the noted ozone peak near a height of 4 m (Fig. 5). The gradient between heights of near 14 m and near 3
m is therefore considered to be a better representation of the in-canopy ozone gradient. Of the 8 short-term gradient
295 measurements (2017), 3 gradients are very similar to the 2019 long-term spring measured gradients (0.14 to 0.32 ppb m$^{-1}$), 3
gradients are near zero (although they are within 1 standard deviation of the long-term spring measurements), and 2 of the
profiles show strong negative gradients (ozone mixing ratio decreasing with height). Near-zero gradients could be caused by
strong mixing. Negative gradients could be due to a change in air-mass above the canopy with a change in winds bringing
cleaner ozone-free air (or, alternatively, plumes of $NO_x$ aloft may decrease ozone aloft, resulting in a decrease of ozone with
300 height). The more frequent positive ozone gradients are consistent with surface-based ozone loss, due to deposition and/or

surface-based chemical losses.

The ozone vertical profiles measured by the tethered-balloon system in 2018 on three measurement days are shown in Figure
7. The measurement was up to a height of 300 m. Each value shown in the figures is an average over a 20-m interval, and the
error bar at each is the standard deviation. The gradient determined from the difference between the highest (300 m) and
305 lowest (20 m) averages ranges from −0.0014 to 0.02 ppb m$^{-1}$, with an average of 0.0087 ppb m$^{-1}$.

The average gradient is approximately 18 times smaller than the afternoon gradient observed in the canopy (0.16 ppb m$^{-1}$
shown in Fig. 6). The positive ozone gradient (i.e., increasing ozone mixing ratio with increasing height) suggests ozone loss
at the surface with continued atmospheric mixing. Comparing the three days of ozone mixing ratios to the wind speeds (not
shown), lower wind speeds are associated with higher ozone mixing ratios over these 3 days, suggesting production and
310 accumulation in the forest. When the wind speed is higher than 3 m s$^{-1}$, lower ozone levels are observed suggesting that the
increase of wind speed dilutes ozone and decreases the ozone mixing ratio, or there may be insufficient time for ozone
production chemistry to occur.

The nearest regular ozone sonde launches are from Stony Plain, more than 400 km SSW of the YAJP site; however, these
launches are rarely done in the afternoon. Between 1986 and 2008, there were 17 launches in the month of June at
315 approximately 17:00 local time (https://woudc.org/). The ozone gradients measured by these sondes between heights of 50 m
and 300 m ranged from −0.0014 to 0.0307 ppb m$^{-1}$, with an average of 0.0081 ppb m$^{-1}$. This average is approximately 7%
smaller than our measured gradient average (0.0087 ppb m$^{-1}$) over the same height range. By comparison, for the month of
July there were 21 launches with an average of 0.0103 ppb m$^{-1}$, approximately 20% greater than our measured gradient
average.

Number: 1    Author:    Subject: Sticky Note    Date: 5/8/2023 9:51:15 PM
Mostly speculation building on a rather superficial understanding of ozone chemistry and vertical mixing.

[revised manuscript text omitted]

---

## Author Comment (AC1)

Note from authors: Reviewer comments are in black text. Our responses are in blue text. Line numbers in the reviewer comments refer to the original manuscript submission, while line numbers in our responses refer to the revised manuscript.

**Response to RC1: 'Comment on acp-2023-26', Anonymous Referee #1, 21 Feb 2023**

This paper uses tower and balloon observations and a 1-D chemical transport model to study ozone over a forest near an region producing oil sand. The manuscript is generally well-written and robust, but a few changes would further strengthen the paper.

The introduction needs some restructuring. The information is rich and almost complete, but poorly connected, thus does not provide the motivation of the study well enough. A paragraph explicitly talking about "Why ozone concentration and deposition over AOSR" is highly recommended.

AC1.1: We thank the reviewer for these comments and feedback. We have reorganized the paragraphs and added text to attempt to improve the connectivity of the information and to better describe the motivation for the study. The structure of the Introduction can now be summarized as the following list of paragraphs…
1) Describe the oil sands.
2) What is ozone and where does it come from.
3) Oher ozone-AOSR studies.
4) Summarize our study motivation (why ozone concentration and deposition over AOSR)
5) Describe how ozone behaves in the canopy.
6) Ozone deposition to the canopy.
7) The effects of turbulence and shading.
8) Study summary.

Text is added to paragraph 4 (lines 36 – 41) as "The motivation for this study is to a) determine how pollutant emissions associated with oil sands extraction modify ozone concentration in the surrounding forest, and b) investigate how the forest affects ozone deposition. A boreal forest site was chosen that is surrounded by oil sands processing facilities including those operated by Syncrude, Suncor, Canadian Natural Resources Limited (CNRL) as well as other facilities. Since exposure to ozone reduces photosynthesis, growth, and other plant functions (Felzer et al., 2007), we investigate what effect the elevated pollution levels of the AOSR have on the surrounding boreal forest and to determine the rate of ozone uptake to the forest."

L 38 – 40: NO does not ALWAYS dominate the in-canopy chemical sink of ozone (e.g. Wolfe et al. (2011) propose BVOC to be the dominant chemical sink in a warm pine forest)

AC1.2: We add the caveat here (Line 51) that "However, Wolfe et al. (2011) have demonstrated that chemical loss due to VOCs can also be significant.".

L 149: Need citation for the observed "shelf shape"

AC1.3: We add Raupach et al., 1996, which is the definitive study as referenced in M17.

Raupach, M.R., Finnigan, J.J. & Brunet, Y. Coherent eddies and turbulence in vegetation canopies. Boundary-Layer Meteorol. 78, 351–382, doi:10.1007/BF00120941, 1996.

L 114: The maximum height of profile measurement was 300m and most of the paper discuss about near-surface turbulent mixing and sinks. Would 1000 vertical layers be an overkill and potentially introducing unnecessary error from vertical transport? Please explain and discuss.

AC1.4: Ozone is assumed constant at the top of the model (at a height of 1 km). This is necessary in order to replace depleted ozone and to simulate downward diffusion from the stratosphere. Ozone at lower heights (near 300 m) would vary diurnally and it would be unrealistic to model it as a constant value. To further explain and discuss our choice of model height we add the following text at Line 199:

"The choice of model height (1001 m) was determined by inspecting vertical ozone profiles from ozonesonde launches at Bratt's Lake (Astitha et al., 2018), which is located approximately 500 km SSW of the oil sands region. The aggregate vertical profile shows a consistently steep gradient between the surface and a height of 1 km (approximately 20 ppb $km^{-1}$) and a much weather gradient between 1 km and 2 km (< 3 ppb $km^{-1}$). Based on this, we choose a 1 km upper boundary of the model and ozone is held constant at this height. Sensitivity to both the assumed constant value and the choice of model height (1001 m) are tested in Section 3.5."

To demonstrate that there is no error introduced by the choice of model height, we run a sensitivity test with a model height of 500 m. The results are added to Table 2 and text is added at Line 504 as "Changing the model maximum height from 1 km to 500 m results in a 30% average overestimation (due to the closer proximity of the canopy-top boundary condition to the measurement height) and a higher RMS error (15.1 ppb); however, the $R^2$ value is slightly improved (0.469 from 0.424)."

Astitha, M., Kioutsioukis, I., Fisseha, G. A., Bianconi, R., Bieser, J., Christensen, J. H., Cooper, O. R., Galmarini, S., Hogrefe, C., Im, U., Johnson, B., Liu, P., Nopmongcol, U., Petropavlovskikh, I., Solazzo, E., Tarasick, D. W., and Yarwood, G.:.Seasonal ozone vertical profiles over North America using the AQMEII3 group of air quality models: model inter-comparison and stratospheric intrusions, Atmos. Chem. Phys., 18, 13925–13945, https://doi.org/10.5194/acp-18-13925-2018, 2018.

L 115, 156 – 157: This approach looks weird, or underexplained at best. In the model, what height does z = 0 correspond to? Assuming z = 0 refers to soil surface, this is not most of the deposition occurs (since leaf surface is mostly the major sink over healthy forest), nor what typical big-leaf model (displacement height) takes. The choice of which layer to put the "big-leaf" foreseeably affect the modelled in-canopy ozone profile. A sensitivity run to explore how the choice of level where the big-leaf is placed, or at least argument for why choosing to put the big-leaf at soil surface is needed.

AC1.5: We acknowledge that this is a significant weakness in our model set-up. Within our model, a flux (or deposition velocity) can only be specified at the model boundary (i.e. the surface). If a deposition surface (or big-leaf) were specified at a high above the ground, there would be deposition to both the underside and the top of that surface. In a future study, we

would like to add deposition at multiple levels as a function of LAI, but this requires a rewriting of the model code that is beyond the scope of this study.

To better explain the operator splitting mechanism of the model we added the following text at Line 152: "This model version uses operator splitting in each minute. First, each species diffuses for 30 seconds using a Crank-Nicholson numerical scheme to solve the diffusion term in Eq. 1. This is followed by 1 minute of uptake or emissions ($E$) and chemistry ($f$). Then the species diffuse for another 30 seconds. This process is repeated 30 times for each 30-min time step."

At Line 192 we add "While a vertical distribution of uptake (or locating the "big leaf" at a specified height above the surface) would be more realistic, this would require placement of the ozone uptake in the emission and chemistry operator step (as opposed to the diffusion operator)."

And finally, text is added at Line 537 in the conclusions as "There is also uncertainty associated with the location of the "big leaf" at the forest floor. In future work, the model could be developed to investigate the effect of a vertical distribution of uptake throughout the canopy height."

L 230: Does GEM-MACH have enough resolution to resolve these regional details?

AC1.6: We add the following text (Line 272) to this discussion:

"The GEM-MACH resolution is 2.5 km and the mines and upgrading facilities are more than 10 km (~4 grid squares) from the tower location. Hence, GEM-MACH can resolve these different sectors within at least ± 7°."

L 310: When the wind speed is higher, there should also be more vertical turbulent mixing generated by horizontal wind shear. This factor should also be considered and discussed in comparing the ozone gradients.

AC1.7: We have added the following text at Line 357: "Higher wind speeds should also be associated with stronger turbulence, due to enhanced wind shear. The stronger turbulence and mixing could lead to a weaker gradient; however, no correlation between the gradient and the wind speed is seen here."

L 393: Does the model explicitly consider the strong diurnal variation of ozone deposition velocity? If not, explain how this might affect your result.

AC1.8:  The model assumes a constant deposition velocity. To discuss this point the following text is added at Line 447: "Rannik et al. (2012) demonstrate a strong diurnal cycle of deposition velocity, averaging 0.2 cm s$^{-1}$ at night compared to more than 0.5 cm s$^{-1}$ during the day (in the summer months). Figure 9b demonstrates that a lower deposition velocity results in a smaller gradient (compare #6, 7, and 8). Although our model unrealistically assumes a deposition velocity that is constant with time, the results suggest that modeling a deposition velocity that is lower at night and a higher during the day would result in a weaker gradient at night and a stronger gradient during the day (relative to a constant value). This would further increase the

difference between the model results and the observations, which show stronger gradients at night."

L 408: How long did snow cover last? Since snow and the compounding low temperature during early season can also significantly reduce ozone dry deposition. This might be a worth-discussing point.

AC1.9: The gradient data are from 27 March to 23 June 2019. Although snow can persist into April in this northern region, snow depth data from the Ft. McMurray airport (Environment Canada historic data) shows that all the snow was melted by 20 March this year. Although some snow may have persisted in the forest due to canopy shading, after 9 April, temperatures were consistently above zero (expect for a few hours at night) with daily highs above 14C for more than 2 weeks.

We have added text at Line 432 as "Although the relative cold in later March and the potential presence of snow might affect the gradients, recalculating the gradients for May and June only results in an average difference ~1% relative to the complete period." While we agree that further investigation of the effects of snow could be an interesting discussion, the colder period seems to have little effect on the observed gradients.

L 464: Direct ozone (and to a lesser extent $NO_2$) flux measurement would also help tremendously to constrain the deposition velocity/flux.

AC1.10: This is an excellent point that we overlooked. We add the text at Line 522: "Ideally, fast ozone (and NO) analyzers could directly measure fluxes (as in Finco et al., 2018) to directly determine deposition velocity and to compare to gradient measurements and deposition parameterizations."
* * *
**Response to RC2: 'Comment on acp-2023-26', Anonymous Referee #2, 09 May 2023**

Ozone chemistry and transport were studied in a boreal forest in the vicinity of oil sands mining in Alberta. Observations rely on gradient measurements within a forest canopy and a few selected tethered balloon profiles. A 1-D canopy model was run with input data from the observations, yielding a comparison between the observation data and the model output. I have no doubt that this project was a challenging effort.

AC2.1: We thank the reviewer for their overview and their appreciation of the challenges of the remote field work, data collection, and modeling.

While the idea to study the influence of emission from the oil sands operations is certainly warranted, the manuscript falls short in producing significant findings that advance the understanding of the chemistry and environmental impacts, or modeling improvements.

AC2.2: We neglected to include important calibration information and statistical analysis of the results and we are grateful to the reviewer for highlighting this and helping to improve the manuscript. We have added statistical analysis to the revised manuscript to emphasize the statistical significance of the findings. This analysis is discussed in more detail below.

What is the primary objective of the study? Investigation of air pollution from oil sands mining? Determining ozone deposition within a boreal forest? Evaluation and advancement of photochemical modeling? The paper provides bits and pieces to these topics, but really only scratches the surface. I don't see a well targeted experimental approach and comprehensive outcomes of this study.

AC2.2: We have expanded on the motivation in the revised Introduction section (see also the response to Reviewer 1, AC1.1). We have added the following text at Line 36: "The motivation for this study is to **a) determine how pollutant emissions associated with oil sands extraction modify ozone concentration in the surrounding forest, and b) investigate how the forest affects ozone deposition**. A boreal forest site was chosen that is surrounded by oil sands processing facilities including those operated by Syncrude, Suncor, Canadian Natural Resources Limited (CNRL) as well as other facilities. Since exposure to ozone reduces photosynthesis, growth, and other plant functions (Felzer et al., 2007), **we investigate what effect the elevated pollution levels of the AOSR have on the surrounding boreal forest and to determine the rate of ozone uptake to the forest.**" (Bold added for emphasis).

There are some fundamental flaws in the experimental approach for determining vertical ozone gradients within the canopy. 2B ozone monitors were used. These instruments are favorable in applications where space and electrical power are limited. However, their analytical performance (precision, accuracy, signal stability (i.e. lack of a drift in the instrument response), sensitivity to water vapor interference, response to monitor temperature changes, etc.) is poorer than for standard ozone monitors. The differences that the authors report in their data for vertical ozone gradients and for ozone transported from different sectors fall well within the measurement uncertainty for these monitors. An additional source of analytical error will likely result from the steady turning on and off of the monitors. The manuscript does not appreciate these challenges and does not present an evaluation of the impact of this operation mode on the instrument performance, stability of its calibration, and gradient data quality. The manuscript contains no information on how instrument response (accuracy) and stability were tracked over time and how gradients were determined to be statistically significant (see for instance [Bocquet et al., 2011]). Statistical analysis are purely data binning and averaging. Because of these reasons most of the interpretation is not warranted and premature.

AC2.3: We thank the reviewer for drawing attention to this oversight. We neglected to provide statistical analysis relevant to the interpretation of the results. We have added a section to provide more information regarding the calibration and the performance of the 2B instruments. We have also calculated confidence intervals for all the data to demonstrate statistical significance.

The added section is as follows (Line 124):

"2.2 Ozone Instrument Uncertainty

The 49i analyzer was laboratory-calibrated prior to the study (a 7-point calibration up to 120 ppb with $R^2 = 0.997$). The standard deviation in the 5-second measurement during calibration was 1.9 ppb. All Model 205 and ozonesonde monitors were calibrated in the field against the 49i analyzer by running the instruments side-by-side for 9 consecutive days period (at a 0.5 Hz frequency) prior to the long-term averaging periods. The ozone mixing ratio varied from near 1.3 ppb to 38 ppb during this period. The root-mean-square errors (RMSE) against the calibrated 49i were less than 2 ppb (at 0.5 Hz). The 2B Ozone Monitor specifications (2B Specifications) give a drift value of < 1 ppb day$^{-1}$ and < 3 ppb year$^{-1}$. Over the 9-day period, the RMSE (at 0.5 Hz) showed no discernable trend with time. A least-squares fit of RMSE with time give a trend of 0.004 ppb day$^{-1}$ (which is not significantly different from zero at a 95% confidence level (C.I.)). Hence, we assume minimal drift during the 3-month measurement period. Water vapour interference is assumed to be minimal since the 2B analyzers have a built-in dryer and heater to eliminate water vapour interference and temperature effects (2B Specifications), and the inlet tubing is only 10 cm in length.

During the long-term, 2019 measurements, when the monitors were activated 4 times per day for one-hour durations, the monitors were allowed to stabilize for 45 minutes and only the last 15-minutes of measurements were used. The manufacturer specifies (2B Specification) a 20-minute warm-up period. The stabilization of the instrument is demonstrated in the supporting information (Fig. S1) from the measured data, indicating full stabilization may require approximately 35 minutes. A truncated mean is calculated from the last 15 minutes, with outliers more than 3 standard deviations from the mean removed (resulting is removal of less than 0.5% of data). The average standard deviation in this 15-minute interval is 1.8 ppb, which gives a 95% C.I. of $\pm 0.24$ ppb (for each 15-minute average)."

Figure 4 is modified to show 95% confidence intervals and the text discussing the comparison (Line 323) is modified as "As discussed in Section 2.2, the 95% C.I. for each 15-mintute average is 0.24 ppb, which would imply a 95% C.I. in each gradient measurement of 0.02 ppb m$^{-1}$. Based on the variability in the long-term measurements, the 95% C.I. of each of the 4 mean gradients is < 0.03 ppb m$^{-1}$. Hence, all measured gradients are significantly different from zero and the overnight/afternoon difference is significant."

The treatise of ozone deposition is lacking understanding of important processes (e.g. deposition to surfaces, humidity effects) and omits important recent findings and literature (e.g. [Clifton et al., 2020]).

AC2.4: We have added text to the last paragraph of the Introduction (Line 71): "A recent review of ozone deposition by Clifton et al. (2020) highlights the need for both short-term field intensives and long-term deposition sites. The review synthetizes the current knowledge of deposition pathways, including stomatal, non-stomatal, and soil uptake and in-canopy chemistry. While our study is not able to distinguish these various pathways, the motivation is to investigate how oil sand extraction and processing affects ozone mixing ratios and deposition."

Discussion of the data is often speculative or depends to a large degree on comparison with other studies, rather than deriving new insight and conclusions from this data set.

AC2.5: With the addition of statistical analysis as suggested by the reviewer (see AC2.3), the conclusions are demonstrated to be significant and no longer speculative.

Understanding and presentation of ozone chemistry is at times superficial.

AC2.6: We assume that this refers to specific points raised in the annotated pdf, which we address below in points AC2.11 to 2.36.

The manuscript presents and builds on observations of nitric oxides and sulfur dioxide but lacks a description of the experiment and data quality procedures.

AC2.7: NO was not observed in this study. This is stated at Line 271: "While NO and $NO_2$ were not measured in this study, we can use GEM-MACH output…". The data shown in Figure 3b are GEM-MACH model data as indicated in the figure, the figure caption, and the text. We believe some poorly worded text discussing measurements made in the Finco et al. (2018) study (which we have corrected) may have caused some confusion and given the wrong impression that we measured NO and/or $NO_2$. The correction is shown in the response to AC2.25 below.

The collection of the $SO_2$ data is described in Section 2.1 at Line 97 "Ozone and $SO_2$ analyzers (49i and 43i, Thermo Scientific) sampled from a height of 2 m." and Line 112 "$SO_2$ measurements (43i, Thermo Scientific and AF22e, Envea) at ground level and a height of 30 m and size-resolved sub-micron aerosol measurements (UHSAS, DMT) helped to further identify wind sectors bringing polluted air to the site.". It is also pointed out in the text and the Fig. 3 caption that the $SO_2$ data presented here are reproduced from the companion paper Gordon et al. (2023). To make this clearer we also add (Line 260) "The $SO_2$ measurements **(reproduced from Gordon et al., 2022)** are from two time periods…" (bold is added text).

The writing/wording is at times inaccurate.

AC2.8: We assume that this refers to specific points raised in the annotated pdf, which we address below in points AC2.11 to 2.36.

I strongly recommend that the modeling work be evaluated by an expert in forest canopy chemistry and modeling.

AC2.9: This comment is for the editor as we have no control over the review process.

An annotated pdf copy with some suggested corrections and more detailed comments will be attached to these summary comments.

AC2.10: We have copied these comments and suggestions into this response and address each comment individually in points AC2.11 to AC2.36.

Annotated pdf corrections.

Line 12: Better wording would be: ...that ozone is destroyed by reaction with nitric oxide emitted from oil and gas extraction operations (as well as NO resulting from photolosis of nitrogen dioxide). emissions). AC2.11: We have changed the wording as suggested.

Line 14: Vertical gradients .... AC2.12: "Vertical" added.

Line 23: of petroleum hydrocarbon extraction from oil sands AC2.13: We change this to "…oil and gas extraction from oil sands…", as we believe this reads easier.

Line 28: , AC2.14: Comma added.

Line 35: and by non-stomatal dry deposition. AC2.15: Text added.

Line 38: AC2.16: "due to the rapid ozone removal" is removed.

Line 40: Poorly worded sentence. AC2.17: Changed to: "The dominant chemical loss process is NO reaction with $O_3$ below the canopy (Kaplan et al., 1988). There are also monodirectional fluxes of NO and $NO_2$, with NO emitted from the soil and $NO_2$ deposited to the ground (Finco et al., 2018)."

Line 55: Poorly worded sentence. AC2.18: We delete "no regional stagnation of air", as this is implied by vertical mixing.

Line 56: Do you mean LAI? AC2.19: We do mean shading and not LAI. The global and regional models discussed in Makar et al. (2017) did not explicitly include the shading of forest canopies in photolysis calculations.

Line 90: Not clear. AC2.20: This is changed to "which was placed 100 m from the tower in the northeast direction (at a wind direction of 40º), since regional winds are typically not from this direction."

Line 93: Need more detailed quality control data for 2B ozone monitors. AC2.21: This is now addressed as described above in AC2.3.

Line 108: Missing quality control and calibration data. AC2.22: Text added "These instruments were factory calibrated and data were quality controlled through visual inspection of the time series, resulting in rejection of less than 0.1% of the data."

Line 181: That seems like a significant omission of an important variable? AC2.23: We are using the Makar et al. (2017) parameterization as described in that paper. We agree that the inclusion of LAI in this parameterization is a good suggestion for future improvement to the Makar et al. parameterization, but addressing this improvement in our manuscript is beyond the scope of this study.

Line 184: How realistic are the isoprene emission rates for this coniferous forest? AC2.24: As stated in the text, these rates are parameterizations for the types of trees in this boreal forest. We do not have data to confirm the Guenther et al. emission rates.

Line 210: NO measurements are not detailed in the measurement section? AC2.25: This was poorly worded as our discussion of the Finco et al. (2018) measurements gave the impression that we are discussing measurement that we made. We add text (Line 250) "While the NO between heights of 5 m and 41 m varied from 0.1 to 2.5 ppb **in the Finco et al. study**, measurements at a height of 0.15 m ranged from 5 to 20 ppb **(Finco et al., 2018)**."

Line 229: Wind sectors are not labeled in Figure 1. AC2.26: We have added the "forest", "polluted", and "other" labels to the figure and changed the figure caption text.

Line 255: Probably none of the sector differences are statistically significant, but are well within the measurement uncertainty and variability of the data. So, this whole discussion about differences should be omitted. AC2.27: We have added discussion of statistical significance (see AC2.3) and have changed the text to reflect this analysis. The final line of this discussion (Line 308) is changed to "Although the results shown here are highly variable, there **appears to be no significant increase in ozone related to increased air pollution.**" (Bold is modified text.)

Line 266: No statistical evaluation of the data. AC2.28: See AC2.3.

Line 268: For radiation measurements perfect leveling of the light sensor is critical. How was that accomplished on a moving pulley? AC2.29: We add the following text "**A table platform was used on the pulley (with ropes attached at each corner) to ensure the UV sensor remained level.** The data were collected for periods of 5-min intervals at 5-m height intervals **after ensuring the sensors were level and not moving.**" (bold text added). Radiation sensors are extremely sensitive to leveling at low light angles, but much less so in summer afternoons when the sun is higher overhead. (i.e., $\sin\theta \approx \sin(\theta + d\theta)$ when $d\theta \ll \theta$)

Line 269: Not a trend. This is a vertical profile. AC2.30: We reword this to "Ozone tended to increase with height when the UV radiation was also increasing with height…".

Line 272: Really not clear if the ozone loss is from reaction with NO or from stromatal uptake or deposition to surfaces and soil. AC2.31: We change "can be explained by" to "could be due to".

Line 273: Really not that obvious. I don't see any obvious consistent ozone profile behavior that stands out of the variability and measurement uncertainty of the data. AC2.32: We have left the sentence in (since there is a peak in many of the profiles), but we add "; however, in many cases, this peak is within the variability of the measurements."

Line 276: There is no plausible reason that ozone should be correlated with radiation as ozone production (dependent on radiation) would happen slower than the time scale of mixing. Further, it's unlikely that there is significant ozone production in this low NOx environment that occurs on time scales that are shorter than the air transport within the domain. AC2.33: This sentence is removed.

Line 300: Mostly speculation building on a rather superficial understanding of ozone chemistry and vertical mixing. AC2.34: This is a list of potential causes of positive or negative vertical gradients. We cannot infer from the reviewer's comment what specifically is wrong with the discussion. We invite the reviewer to provide further information detailing what elements of this explanation are superficial.

Line 369: NO will show very significant diurnal cycles. I can not see how any modeling assuming a constant level of NO is going to produce realistic results. AC2.35: We agree that diurnal variation in NO would be more realistic. However, the use of a constant NO produces better model agreement. Since this is already discussed in the manuscript, it is not clear from the reviewer's comment how this should be addressed further. We invite specific information outlining what corrections are required.

Line 465: The study lacks a clear analysis of the turbulent measurements within and above the canopy. Without those data there really isn't a good way for simulating the vertical transport and chemistry within the canopy AC2.36: Measured values within and above the canopy are used to drive the model. This is outlined in Section 2.2. It isn't obvious to us what the reviewer means by a lack of clear analysis of the turbulent measurements. We would invite a more detailed explanation of what exactly is lacking in the analysis.

References:

Bocquet, F., D. Helmig, B. A. Van Dam, and C. W. Fairall (2011), Evaluation of the flux gradient technique for measurement of ozone surface fluxes over snowpack at Summit, Greenland, Atmospheric Measurement Techniques, 4, 2305-2321, doi:10.5194/amt-4-2305-2011.

Clifton, O. E., A. M. Fiore, W. J. Massman, C. B. Baublitz, M. Coyle, L. Emberson, S. Fares, D. K. Farmer, P. Gentine, G. Gerosa, A. B. Guenther, D. Helmig, D. L. Lombardozzi, J. W. Munger, E. G. Patton, S. E. Pusede, D. B. Schwede, S. J. Silva, M. Sorgel, A. L. Steiner, and A. P. K. Tai (2020), Dry Deposition of Ozone Over Land: Processes, Measurement, and Modeling, Reviews of Geophysics, 58, doi:10.1029/2019rg000670.

---

## Referee Report (RR1)

**COMMENTS TO THE AUTHOR(S)**

Zhang et al. and coauthors study ozone temporal and spatial variations in a boreal forest using observational data and a 1D canopy model. They sought to understand the influences of the oil sand extraction on ozone concentration in the forest and to investigate the sensitivity of modeled ozone concentrations to different in-canopy processes especially dry deposition. The primary findings of the study are that (i) there are no significant changes in ozone levels due to oil sands extraction; (ii) modeled ozone vertical gradients are highly sensitive to NO schemes, vertical mixing, and dry deposition.

This paper pursues an interesting topic that in dire need of development using high spatial resolution canopy model. However, the paper feels incomplete. There are several significant issues relating to the level of details and the numerical experiments. On the observation–the long-term spring/early summer ozone data suggest no significant ozone increases, while short-term summertime ozone data show either higher or lower ozone. I find the interpretation of these results is lacking. What leads to the higher ozone at the forest site during Jun 9-18 2018? What is responsible for the lower ozone during 10 Jun-15 Aug in 2018? Would you expect the spring ozone behave the same as summertime ozone (i.e., no significant changes in ozone in summer as well) and why? In addition, information about the anthropogenic emissions of ozone precursors (NOx and VOCs) from the nearby industry could help interpret the results but again is missing. On the modeling–I appreciate the efforts the authors take to set up the 1D canopy model. However, it was hard for me to find the novelty and implications in terms of forest canopy modeling. Modeling the nighttime concentration gradients has been difficult due to challenges in micrometeorological measurements and the K-theory has been known for its inadequacy in representing nighttime vertical mixing. Therefore, the conclusion on vertical mixing does not seem new to me. The sensitivity tests are helpful to understand how the model works, but I feel some of the details do not need to be written out–for example, there are overlaps in section 3.4 and 3.5 in terms of testing deposition velocity and vertical mixing. The modeling work on dry deposition (i.e., assuming a constant velocity and corresponding sensitivity analysis) seems inadequate to answer the second research question raised in the manuscript (i.e., how the forest affects ozone deposition).

On presentation and details:

The introduction still lacks the background review of literature necessary to put this study in context. Some terminology is inaccurate and not consistent throughout the manuscript, for instance, "diffusion" and "vertical mixing", "deposition" and "dry deposition". Some sentences need further clarification (Please see line-by-line comments). Some figures are hard to read (Please see line-by-line comments).

In summary, the paper has an interesting premise. I believe it takes immense efforts to set up the model. But I feel it is half-baked.

**Line by line comments:**

Line 11 "as much as 10 ppb lower". This seems contradictory with the conclusion that "no significant increase in ozone levels". Please reconcile.

Lines 12-13 "This finding is supported by…". I would focus on the results from this study in the Abstract.

Line 37 "b) investigate how the forest affects ozone deposition". I think the results show how (presumably dry) deposition impacts modeled ozone concentrations but provide little answers to how the forest affects ozone deposition. In addition, line 74 "how oil sand … affects … and deposition", it seems a completely different research question than the one in line 37.

Lines 42-59: Ozone (dry) deposition is discussed here. And at the end of the introduction (lines 71-73), the authors went back to dry deposition. It seems disconnected for me and as a result, the research questions are not put into context for readers.

Lines 60-70: Are the statements here based on the one paper Makar et al. (2017)? If so, this can be more concise. If not, please add references for the statements such as "Including both …97%..." and etc.

Line 148: "diffusion" is ambiguous. It should be "turbulent diffusion" or "vertical mixing" or "turbulent mixing". Please pick one terminology and be consistent throughout the manuscript.

Line 150 equation (1): I would keep the subscripts consistent. If subscript n is preferred, I would change $K(Z_n)$ to $K_n$. If the parenthesis is preferred, I would use $C_m(z)$, $E_m(z)$, and $K(z)$. $K(Z_n)$ doesn't really make sense to me and it is not consistent with other variables.

Line 152: these modifications should be reflected in the equation.

Line 155 "This process is repeated 30 times for each 30-min time step". Do you mean the timestep for the model run is 1 min? Please clarify.

Lines 159-162 "Initially, the temperature … (Section 3.5)". Running with a constant temperature above the canopy does not make sense because temperature decreases adiabatically. The sensitivity test with a constant temperature is not necessary because it does not happen in the atmosphere. In addition, I would specify "air temperature" here because some canopy models calculate leaf temperature too.

Line 168: "GEM-MACH". Please explain what model it is and why the K values form this model is applicable here.

Line 171 "diffusion coefficient". Ambiguous because it can mean molecular diffusion. Normally K is referred to as "eddy diffusivity".

Line 179: I don't remember the eddy covariance system(s) are mentioned in the Methods section. If you used the data in the manuscript, please add the instrumentation to Section 2.1.

Lines 271-274: I would move it to the Methods section. It breaks the flow of results here.

Lines 286-287 "This indicates that the NOx… more significant photochemical aging". Why is NOx from other directions more aged?

Lines 288-290 "While… superimposed". I find it rather confusing.

Line 305 "not statistically different…" Can you show the statistics? In addition, I think you suspected NO titration at night and I am guessing you think it is the reason why ozone in the polluted wind sector is lower? If that's the case, can you explain this point more clearly in the text because it is very not obvious to me. If not, can you explain the possible chemical and physical processes behind the results?

Line 310 Section 3.2: it is not clear to me what main results are for this section. I think it is really hard to explain the results and extract important information without presenting turbulence data (such as sigma_w). I would recommend thinking of what new results you get from the in-canopy profiles in terms of vertical mixing and deposition and focus on them, instead of describing each figure.

Line 367 "3.3 Modeling Comparison". It is ambiguous. I would clarify that it is compared with measurements of diurnal cycle of ozone concentration above the canopy.

Lines 368-376: I would move the text to the Methods section.

Line 393 "removing deposition…" I don't think this a viable experiment design because it is unrealistic assuming no dry deposition. In addition, I think the manuscript investigates "ozone dry deposition". Please keep the terminology consistent and accurate.

Line 395 "although diffusion …" new paragraph.

Line 408-409 "The worse aspect of the model behaviour…" It sounds like an "entrainment" problem to me.

Line 417 "Gradient Comparison". I would add "vertical gradient" at least.

Section 3.4 I am not sure that deltaZ is a reliable metric to evaluation the model performance because it misses out a lot of information such as the absolute values and all the layers in between. Can you justify that this is a good metric to do model evaluation in terms of vertical gradients? I would just add the model results to Figure 5&7 to do the evaluation.

Section 3.5 Additional Sensitivity Analysis. I feel this section can be integrated into other sensitivity analysis. Also, is this compared with observed diurnal cycle or vertical profiles? I would reorganize all the sensitivity analysis based on different processes. The manuscript looks a bit disorganized right now.

Again, lines 479-493 can be moved to the Methods section.

Line 503 "isoprene emission rate". I don't think emission algorithm is mentioned in the model description. Is it based on leaf or air temperature? Is it calculated at each canopy layer or just like the dry deposition assuming a "big leaf"?

Figure 3, just curious why CO2 is plotted as a staircase plot and others dot?

Figure 4d: why ozone is higher at night?

Figure 5: I would change the y axis to z/h. z is height and h is the canopy height.

Figure 8. This figure is really hard to read. I would separate them into panels. They can be grouped into (i) base case and OBS; (ii) NO cases and OBS; (iii) vd cases and OBS; (iv) K cases and OBS.

Figure 9: again, I am not sure if deltaZ is the best metric to evaluate the model.

---

## Author Response (AR2)

Note from authors: Reviewer comments are in black text. Our responses are in blue text. Line numbers in the reviewer comments refer to the original manuscript submission, while line numbers in our responses refer to the revised manuscript.

**Response to RC3: 'Comment on acp-2023-26', Anonymous Referee #3, 20 June 2023**

Zhang et al. and coauthors study ozone temporal and spatial variations in a boreal forest using observational data and a 1D canopy model. They sought to understand the influences of the oil sand extraction on ozone concentration in the forest and to investigate the sensitivity of modeled ozone concentrations to different in-canopy processes especially dry deposition. The primary findings of the study are that (i) there are no significant changes in ozone levels due to oil sands extraction; (ii) modeled ozone vertical gradients are highly sensitive to NO schemes, vertical mixing, and dry deposition.

AC3.1: We thank the reviewer for these comments and feedback. As part of the revisions to the manuscript we have tried to highlight the conclusions more clearly. The primary conclusions from the study are that (i) there are no significant changes in ozone levels due to oil sands extraction, and (ii) the model results suggest a deposition velocity between 0.2 cm s$^{-1}$ and 0.4 cm s$^{-1}$. Although we do conclude that modeled ozone vertical gradients are highly sensitive to NO schemes, vertical mixing, and dry deposition, these are not the primary conclusions of the study. However, these results are included in the Conclusions section since they would be useful for further modeling studies.

To make the motivation clearer, we have modified the text at line 36 to "The motivation for this study is to a) determine how pollutant emissions associated with oil sands extraction modify ozone concentration in the surrounding forest, and b) **estimate the dry deposition velocity of ozone to the surrounding forest**" (modified text in bold). Additionally at line 79, the text is modified as "the motivation is to investigate how oil sand extraction and processing affects ozone mixing ratios and **to determine the total dry deposition velocity at this location**."

We also wish to point out that while investigating a reviewer comment (AC3.39), we discovered an error in our analysis code that has changed the results shown in the lower panel of Fig. 4 (so that measured ozone is lower at night, as would be expected). This has also slightly changed the diurnal variation of the gradient (Figs. 6 and 9b). This has resulted in some modification to the discussion, and we conclude that the model infers a deposition velocity in the range of 0.2 to 0.4 cm s$^{-1}$ for this location. We apologize for not noticing the error sooner and thank the reviewer for pointing it out.

This paper pursues an interesting topic that in dire need of development using high spatial resolution canopy model. However, the paper feels incomplete. There are several significant issues relating to the level of details and the numerical experiments. On the observation–the longterm spring/early summer ozone data suggest no significant ozone increases, while short-term summertime ozone data show either higher or lower ozone. I find the interpretation of these results is lacking. What leads to the higher ozone at the forest site during Jun 9-18 2018? What is responsible for the lower ozone during 10 Jun-15 Aug in 2018? Would you expect the spring

ozone behave the same as summertime ozone (i.e., no significant changes in ozone in summer as well) and why?

AC3.2: We believe that the difference between short-term and long-term averages may be due to tropospheric folding events.  The following text is added at Line 342:

"Hence, the long-term diurnal averages (separated by sector) suggest no significant ozone increases associated with industrial pollution, while short-term summertime ozone data shows inconsistent results, with either higher or lower ozone from the industrial sector. The impact of tropospheric folding events, known as stratospheric intrusions, can impact ozone mixing ratios at the surface (Pendlebury et al. 2018). Other work (Makar et al, 2023) shows a high correlation between monthly ozone averages and the number of stratospheric ozone exchange events occurring within each month, the latter detected by ozone LIDAR within 20km of the site (Makar et al, 2023).  These events have been shown to contribute an additional 10 ppbv to monthly average ozone relative to the ambient atmosphere prior to the events. Since these events happen at varying frequencies with time scales on the order of 1 week, they provide a likely cause of higher ozone over short periods, while over longer periods the effects of the intrusions would be averaged out.  Lidar measurements outlined in Makar et al. (2023) demonstrate a stratospheric intrusion in the AOSR on 6 to 7 Jun 2018, which likely modified the ozone mixing ratios during the 10-day 9 to 18 Jun 2018 period, resulting in more variability and higher concentrations relative to the longer measurement periods. Since intrusion frequencies are relatively constant between Jan to Jun (and less frequent in late summer), we do not expect the time intervals of the different measurement periods to have a significant effect on the ozone differences between sectors."

In addition, information about the anthropogenic emissions of ozone precursors (NOx and VOCs) from the nearby industry could help interpret the results but again is missing.

AC3.3: The topic of emissions in the region has been published in previous work (Zhang et al., 2018); we have added a summary of emissions data from Zhang et al. (2018) at Line 97 as follows:

"Emissions in the region are summarized in Zhang et al. (2018), which reviewed national, provincial, and local emissions inventories for the Oil Sands Region between 2010 and 2013 (up to 7 years prior to the start of this study). Zhang et al. (2018) report annual totals of 18,000 t (tonnes) CO, 39,600 t NOx, 1,000 t PM2.5, 1,100 t PM10, 760 t SO$_2$, and 34,000 t VOCs. More than 40% of the CO, PM2.5, and PM10, emissions were from the Suncor facility (Fig. 1), while nearly 50% of the SO$_2$ and VOC emissions were from the Syncrude facility (Fig. 1). Significant VOCs (> 1000 t/a) included higher alkenes, higher alkanes, higher aromatics, propane, isoprene, and toluene."

We would also like to reference the test case results shown in the figure below for the reviewer's information – these are not included in the revised manuscript as the modelling work is in development. The figure below shows the difference in surface-level ozone between two 2.5km horizontal resolution GEM-MACH model runs: a "base case" including all industrial emissions, and a "scenario" in which all oil sands facility emissions are removed.  The figure shows the

difference (base case – scenario) for ground level ozone. Positive (red) colours indicate areas where ozone is higher in the base case than in the scenario (i.e. ozone that may be attributed to photochemical production). Negative (blue) colours indicate areas where the ozone is lower in the base case than in the scenario (i.e. regions where titration of ozone by NOx results in lower ozone concentrations, and hence when oil sands emissions are removed, the ozone concentration increases). The titration-driven reduction in ozone due to industry emissions is more than 2 ppb near the emission sources, near 1 ppb at the YAJP tower location (red dot on right panel). That is, the tower is placed within the region where NOx titration of ozone is dominating the local ozone budget. Further from the NOx sources (beyond 50 to 100 km), the contribution of oil sands emissions towards ozone production results in ozone concentration increases of 0.1 to 0.3 ppbv (note that the concentration difference colour scale is logarithmic). This supports our hypothesis that removal of ozone by NO$_x$ titration is a dominant process at the YAJP tower.

[Figure]

On the modeling–I appreciate the efforts the authors take to set up the 1D canopy model. However, it was hard for me to find the novelty and implications in terms of forest canopy modeling. Modeling the nighttime concentration gradients has been difficult due to challenges in micrometeorological measurements and the K-theory has been known for its inadequacy in representing nighttime vertical mixing. Therefore, the conclusion on vertical mixing does not seem new to me.

AC3.4: As we state above (AC3.1), our findings regarding overnight mixing are not the primary conclusion of the work, though we note that our work supports previous work. Our main intent with the work and the model was the use of the model to estimate the dry deposition velocity for this forest. To make this clearer, we modify the text at line 555 as "The reduced overnight mixing may suggest that modeling nighttime stability using the Obukhov length (Eq. 2) does not account for the increased stability within a canopy associated with canopy decoupling, **which further demonstrates a known weakness in using a local gradient-diffusion model (K-theory) to model nighttime canopy mixing (e.g., Lee and Mahrt, 2005).**" (added text in bold).

The sensitivity tests are helpful to understand how the model works, but I feel some of the details do not need to be written out–for example, there are overlaps in section 3.4 and 3.5 in terms of testing deposition velocity and vertical mixing.

AC3.5: We agree that including both the list of configurations and the sensitivity test results in some overlap and confusion, as some of the configurations could be considered sensitivity tests. We make the distinction that the "configurations" are used to modify the model to improve model-observation agreement, whereas the "sensitivity tests" are done to demonstrate that the model output is not overly sensitive to the choice of model parameters. We have clarified the manuscript by adding a paragraph at the end of Section 2.3 (See also AC3.28) as:

"A series of model configurations were chosen to investigate different physical mechanisms and their potential effect on the diurnal variation of ozone mixing ratios and the gradients above and within the canopy and to improve the measurement to model comparison. These model configurations are listed in Table 1. The model was run for each configuration for the period from 18:00 (local) 20 June to 06:00 (local) 27 July 2018. We disregard the first 12 hours for model spin-up, resulting in 6 days of model output. The first 5 configurations are variations in input NO, discussed in the following section. Configurations #6-8 vary the ozone deposition velocity to 0 (#6), 2 cm s$^{-1}$ (#7), and 0.8 cm s$^{-1}$ (#8) (from the base case of 0.4 cm s$^{-1}$). Although it is unrealistic to assume no deposition of ozone, this configuration was included as a demonstration of the extent to which the gradient depends on deposition alone. Configurations #9 and #10 vary the strength of turbulent mixing by a factor of 0.5 and 2 respectively (at all heights). To compare model output and measurements, the 10-min measurements at a height of 22 m were averaged to 30-min values."

The text at the start of Section 3.2 is modified to:

"The modeled ozone mixing ratio for each configuration **listed in Table 1** is compared to measured values (both at heights of 22 m) in Figure 8. Statistics (ratio of modeled to observed averages, RMS error, and $R^2$) for the runs are listed in Table 1."

Since the Conclusions section discussed only the results of the configurations, we have also moved the sensitivity test from the main text of the manuscript to the supporting information (Text S1 and Table S1 in the revised manuscript). This should allow a reader interested in the mechanics of the model to investigate further, while not distracting from the main results of the study.

The modeling work on dry deposition (i.e., assuming a constant velocity and corresponding sensitivity analysis) seems inadequate to answer the second research question raised in the manuscript (i.e., how the forest affects ozone deposition).

AC3.6: As discussed in AC3.1, the second research question was poorly worded and has been rephrased as "estimate the dry deposition rate of ozone to the surrounding forest". The model is used to estimate the dry deposition rate for this forest.

On presentation and details:

The introduction still lacks the background review of literature necessary to put this study in context.

AC3.7: We have added a paragraph at line 52 discussing a relevant paper which was overlooked:

"Clifton et al., 2021 used a large eddy simulation coupled to a multilayer canopy model to investigate ozone removal by a deciduous forest. They found that organized turbulence leads to heterogenous mixing which can slow down or speed up reaction rates of ozone at different heights in the canopy. They found low covariance between ozone mixing ratio and leaf uptake (due to the effects of organized turbulence). This finding effectively questions the use of a deposition velocity in estimating ozone fluxes, since the uptake flux of ozone is proportional to the ozone mixing ratio for a given deposition velocity. Nevertheless, the analysis also suggests that organized turbulence does not likely bias estimates of ozone dry deposition during summertime afternoon conditions."

We also add at line 41: "The importance of correctly modeling dry deposition to the forest is demonstrated by Clifton et al. (2020b), who find that variation in deposition schemes leads to mean summertime biases of −4 to 7 ppb. A review of ozone deposition velocity schemes used in current models may be found in Clifton et al (2023)."

Beyond this, without specific indication of what is missing from the review, we are not aware of what else is missing. We invite further comment from the reviewer to expand on what is required.

Some terminology is inaccurate and not consistent throughout the manuscript, for instance, "diffusion" and "vertical mixing", "deposition" and "dry deposition".

AC3.8: We have changed "deposition" to "dry deposition" where appropriate throughout the manuscript. "Diffusion" has been changed to "turbulent mixing" or "vertical turbulent mixing" and the "diffusion coefficient" is changed to "eddy diffusivity".

Some sentences need further clarification (Please see line-by-line comments). Some figures are hard to read (Please see line-by-line comments).

In summary, the paper has an interesting premise. I believe it takes immense efforts to set up the model. But I feel it is not fully thought through.

Line by line comments:

Line 11 "as much as 10 ppb lower". This seems contradictory with the conclusion that "no significant increase in ozone levels". Please reconcile.

AC3.9: The text in the abstract is changed to "there is no significant increase in ozone mixing ratio…".

Lines 12-13 "This finding is supported by…". I would focus on the results from this study in the Abstract.

AC3.10: "This finding is supported by previous studies which suggest that…" is modified to "This suggests that…".

Line 37 "b) investigate how the forest affects ozone deposition". I think the results show how (presumably dry) deposition impacts modeled ozone concentrations but provide little answers to how the forest affects ozone deposition. In addition, line 74 "how oil sand … affects … and deposition", it seems a completely different research question than the one in line 37.

AC3.11: See AC3.1. The wording has been revised to indicate that our main purpose in the work was to estimate the average ozone dry deposition velocity to the forest.

Lines 42-59: Ozone (dry) deposition is discussed here. And at the end of the introduction (lines 71-73), the authors went back to dry deposition. It seems disconnected for me and as a result, the research questions are not put into context for readers.

AC3.12: The discussion of the Makar et al. paper (which was originally between lines 59 and 71) is moved to follow the student motivation. Now the flow of the discussion moves from the study motivation (lines 36-43), followed the discussion of the importance of canopy shading and turbulence in modeling ozone, followed by paragraphs discussing dry deposition.

Lines 60-70: Are the statements here based on the one paper Makar et al. (2017)? If so, this can be more concise. If not, please add references for the statements such as "Including both …97%..." and etc.

AC3.13: To make this paragraph more concise and to make the attribution clear, it is reduced as follows…

"Makar et al. (2017), herein M17, demonstrated that the turbulence and shading effects of forests on ozone mixing and chemistry have been poorly modeled in global and regional air-quality models. They found that including both these effects in a regional air-quality model accounted for 97% of the previous positive bias in forested regions. Approximately one-third of this improvement was attributed to the shading effect, while two-thirds was due to the change in turbulence parameterization. Hence, this paper suggests that any accurate modeling of ozone within a forest must include both turbulence and shading effects. Testing currently underway with the CMAQ air-quality model supports these results, showing a significant improvement in model surface ozone biases when these effects are included (Campbell et al, 2021)."

Line 148: "diffusion" is ambiguous. It should be "turbulent diffusion" or "vertical mixing" or "turbulent mixing". Please pick one terminology and be consistent throughout the manuscript.

AC3.14: Done (See AC3.8).

Line 150 equation (1): I would keep the subscripts consistent. If subscript n is preferred, I would change K(Zn) to Kn. If the parenthesis is preferred, I would use Cm(z), Em(z), and K(z). K(Zn) doesn't really make sense to me and it is not consistent with other variables.

AC3.15: Done. (While this breaks from the convention used in three previously published papers that use this model, we see no issue in changing it here.)

Line 152: these modifications should be reflected in the equation.

AC3.16: The inclusion of sesquiterpenes adds more output species but doesn't change the equation. Similarly, changing the diffusion code to a Crank-Nicholson scheme only changes how the equation is solved. The surface deposition is modeled as negative emissions, which was not explained well in the text. We modify the following lines at 168 and 174:

"In the 1-D canopy model, the rate of change of each chemical species mixing ratio ($C$) at each model level is calculated due to their emissions **or uptake** (**positive or negative** $E$, **respectively**), chemical reactions ($f$) and diffusion (Eq. 1) at each layer."

"Deposition is added as uptake (a negative mass rate of change E) at the lowest level."

Line 155 "This process is repeated 30 times for each 30-min time step". Do you mean the timestep for the model run is 1 min? Please clarify.

AC3.17: The text is modified as:

"This model version uses operator splitting in each minute. First, each species diffuses for 30 seconds **(with a time step of 1 second)** using a Crank-Nicholson numerical scheme to solve the turbulent mixing term in Eq. 1. This is followed by 1 minute of uptake or emissions (E) and chemistry (f). Then the species diffuse for another 30 seconds. **The operator splitting** process is repeated 30 times for each 30-min **output** time step."

Lines 159-162 "Initially, the temperature … (Section 3.5)". Running with a constant temperature above the canopy does not make sense because temperature decreases adiabatically. The sensitivity test with a constant temperature is not necessary because it does not happen in the atmosphere. In addition, I would specify "air temperature" here because some canopy models calculate leaf temperature too.

AC3.18: We have replaced occurrences of "temperature" with "air temperature". "Initially" in this sentence refers to model runs for previous studies and in hindsight this isn't relevant here. We rephrase this as

"The air temperature above a height of 29 m was modeled assuming a dry adiabatic lapse rate ($0.0098$ K m$^{-1}$) above this height."

We disagree with the assertion that the sensitivity test is not necessary because it is unrealistic. It is useful to demonstrate model sensitivity using extreme examples, especially when those tests demonstrate very little effect on the model results – as is the case here (see Table S1).

Line 168: "GEM-MACH". Please explain what model it is and why the K values form this model is applicable here.

AC3.19: Because of the modification discussed in AC3.22, GEM-MACH is now introduced and discussed in Section 2.1. The added text (at line 138) is:

"For chemical species not measured in this study (NO, NO$_2$, and eddy diffusivity $K$), we use output from the GEM-MACH model (Global Environmental Multiscale-Modeling Air-Quality and Chemistry), which is the regional chemical transport model used by Environment and Climate Change Canada. GEM-MACH has been used for numerous modeling studies focussed on the AOSR (e.g. Makar et al., 2018; Whaley et al., 2018; Fathi et al., 2021). The model provides turbulence parameters which are consistent with meteorological forecasts for the region."

Line 171 "diffusion coefficient". Ambiguous because it can mean molecular diffusion. Normally K is referred to as "eddy diffusivity".

AC3.20: Done (See AC3.8).

Line 179: I don't remember the eddy covariance system(s) are mentioned in the Methods section. If you used the data in the manuscript, please add the instrumentation to Section 2.1.

AC3.21: Sonic anemometers are mentioned in Section 2.1 at line 129.

Lines 271-274: I would move it to the Methods section. It breaks the flow of results here.

AC3.22: These lines are moved to the end of Section 2.1 (line 138-144) and rephrased as:

"For chemical species not measured in this study (NO, NO$_2$, and eddy diffusivity $K$), we use output from the GEM-MACH model... The GEM-MACH resolution is 2.5 km and the mines and upgrading facilities are more than 10 km (~4 grid squares) from the tower location. Hence, GEM-MACH can resolve source locations within at least ± 7°."

The text at line 308 is modified to "NO and NO$_2$ are included in the comparison using GEM-MACH output for the period between 1 Jun and 17 Aug (Fig. 3b)."

Lines 286-287 "This indicates that the NOx... more significant photochemical aging". Why is NOx from other directions more aged?

AC3.23: Text added: "…, likely due to more distant sources or eventual recirculation of oil sands emissions."

Lines 288-290 "While… superimposed". I find it rather confusing.

AC3.24: Changed to "While our companion paper (Jiang et al., 2022) describes source locations for aerosols with finer angular resolution, this is not possible with the ozone measurements since ozone **mixing ratio also varies by time of day**. There are not enough data to separate both wind direction and time-of-day into more than 3 sectors."

Line 305 "not statistically different…" Can you show the statistics? In addition, I think you suspected NO titration at night and I am guessing you think it is the reason why ozone in the polluted wind sector is lower? If that's the case, can you explain this point more clearly in the text because it is very not obvious to me. If not, can you explain the possible chemical and physical processes behind the results?

AC3.25: Text added following line 340 as "… although these values are not statistically different from the polluted sector **(as demonstrated by the overlap of the 95% CI in Figure 4)**". Text is added at line 358 as "As with both Cho et al. (2017) and Aggarwal et al. (2018), we hypothesise that this it due to ozone titration by NO."

Line 310 Section 3.2: it is not clear to me what main results are for this section. I think it is really hard to explain the results and extract important information without presenting turbulence data (such as sigma_w). I would recommend thinking of what new results you get from the in-canopy profiles in terms of vertical mixing and deposition and focus on them, instead of describing each figure.

AC3.26: We agree that this was not clear, and we regret not having measurements of turbulence profiles, which would have helped the discussion and interpretation. We have added the following at the start (line 362) and end (line 420) of the section:

"Understanding the vertical variation of ozone in and above the canopy is necessary since we use the comparison of measured and modeled gradients to infer deposition velocity. Here we describe vertical profile measurements of ozone and compare these measurements to other studies."

"Hence, these measurements demonstrate the substantial variability of the short-term, in-canopy vertical profiles, which are affected by shading. The longer-term measurements show a clear diurnal variation in the gradient, with a weaker gradient in the morning and stronger gradients overnight. Above-canopy gradients in the afternoon show less variability and are relatively consistent with gradients derived from ozone sonde measurements outside the region."

Line 367 "3.3 Modeling Comparison". It is ambiguous. I would clarify that it is compared with measurements of diurnal cycle of ozone concentration above the canopy.

AC3.27: Changed to "Above-canopy Ozone Mixing Ratio Comparison".

Lines 368-376: I would move the text to the Methods section.

AC3.28:  See AC3.5.

Line 393 "removing deposition…" I don't think this a viable experiment design because it is unrealistic assuming no dry deposition. In addition, I think the manuscript investigates "ozone dry deposition". Please keep the terminology consistent and accurate.

AC3.29: As discussed in AC3.18, we do not believe an unrealistic model condition demonstrated for comparison is not useful or viable; it helps determine the relative importance of a process. Here, leaving ozone deposition completely out of the model shows, through comparison to the other simulations, the extent to which deposition may alter the vertical profile of ozone. The text is added at line 259 (and also shown above in AC3.28) as "Although it is unrealistic to assume no deposition of ozone, this configuration is included as a demonstration of how dependent the gradient is on deposition alone."

Line 395 "although diffusion …" new paragraph.

AC3.30:  Changed.

Line 408-409 "The worse aspect of the model behaviour…" It sounds like an "entrainment" problem to me.

AC3.31:  While we believe the reviewer is likely correct, we'd prefer not to speculate here.

Line 417 "Gradient Comparison". I would add "vertical gradient" at least.

AC3.32:  We have changed the Section title to "Above- and In-Canopy Vertical Gradient Comparison".

Section 3.4 I am not sure that deltaZ is a reliable metric to evaluation the model performance because it misses out a lot of information such as the absolute values and all the layers in between. Can you justify that this is a good metric to do model evaluation in terms of vertical gradients? I would just add the model results to Figure 5&7 to do the evaluation.

AC3.33: We have added model results to Figures 5 and 7 for comparison. The 2-point gradient measurements ($\Delta[O_3]/\Delta z$) are available over a 3-month period, 4 times a day.  The in-canopy and above-canopy profiles were measured over 5 and 3 days, respectively with only 2 to 4 profiles measured in each day.  As described in Section 3.2 and shown in Fig 6 there is substantial variability in the profile measurements, whereas there are enough 2-point gradient measurements to give significant differences in the diurnal variation.

The following text is added at the start of Section 3.4 (line 468): "Modeled vertical ozone profiles are compared to the measured tethersonde profiles in Figure 7. These hourly profiles (shown for 4 hours at the same time of day as the measurements) are averages for the 6 days of the model run. The modeled vertical profiles demonstrate nearly linear vertical gradients in the 20 to 300 m range. This range is used to calculate the 2-point gradient $d[O_3]/dz$) shown in Figure 9."

And at line 482: "Modeled vertical profiles within the canopy are compared to the measured vertical profiles in Figure 5. Stronger curvature is seen in these profiles in the upper half of the canopy (see also the below-canopy profiles in Figure 7). To compare the model profiles to the long-term measurements, the gradient between 2 m and 25 m is calculated from the model output."

Section 3.5 Additional Sensitivity Analysis. I feel this section can be integrated into other sensitivity analysis. Also, is this compared with observed diurnal cycle or vertical profiles? I would reorganize all the sensitivity analysis based on different processes. The manuscript looks a bit disorganized right now.

AC3.34: We have reorganized by moving the sensitivity analysis to the supporting information section. (See AC3.5).

Again, lines 479-493 can be moved to the Methods section.

AC3.35: This section is moved to the supporting information section. (See AC3.5).

Line 503 "isoprene emission rate". I don't think emission algorithm is mentioned in the model description. Is it based on leaf or air temperature? Is it calculated at each canopy layer or just like the dry deposition assuming a "big leaf"?

AC3.36: The algorithm is referenced at line 247 as "As is outlined in Makar et al. (1999), the model includes forest emissions of isoprene and monoterpenes following Guenther et al. (1993).".  We add the text "The emission rates are functions of air temperature (used as a proxy for leaf temperature) and LAI and emissions are at each model layer in the canopy."

Figure 3, just curious why CO2 is plotted as a staircase plot and others dot?

AC3.37:  The $CO_2$ gas analyser was a permanent installation over 4 years (see line 129-133). Hence there are too much data for a scatter plot. We could add an explanation if required, but it might needlessly interrupt the discussion of the results.

Figure 4d: why ozone is higher at night?

AC3.38: As discussed in AC3.1, we investigated this question and discovered an error in the analysis code. The ozone levels are now much lower at night as would be expected.  We thank the reviewer again for drawing attention to this so that we could discover our error.

Figure 5: I would change the y axis to z/h. z is height and h is the canopy height.

AC3.39: The right axes of figures 5 and 7 have been modified to show the heights relative to the canopy height. Now both real height in meters and relative height are available on each figure.

Figure 8. This figure is really hard to read. I would separate them into panels. They can be grouped into (i) base case and OBS; (ii) NO cases and OBS; (iii) vd cases and OBS; (iv) K cases and OBS.

AC3.40: We have modified the figure, but we instead use two groups including a) OBS + base + NO cases, and b) OBS + base + vd and K cases.  We have also done the same expansion to Figure 9b. While we can appreciate the suggested 4 panels, many of the scenarios overlap and further separation doesn't seem necessary.

Figure 9: again, I am not sure if deltaZ is the best metric to evaluate the model.

AC3.41: See AC3.33.